# The "ABC-DA system" (v1.4): a variational data assimilation system for convective scale assimilation research with a study of the impact of a balance constraint

Ross Noel Bannister[1]

[1]National Centre for Earth Observation, Department of Meteorology, University of Reading, Reading, RG6 6BB, UK
**Correspondence:** Ross Bannister (r.n.bannister@reading.ac.uk)

**Abstract.** Following the development of the simplified atmospheric convective-scale 'toy' model (the ABC model, named after its three key parameters: the pure gravity wave frequency, $A$, the controller of the acoustic wave speed, $B$, and the constant of proportionality between pressure and density perturbations, $C$), this paper introduces its associated variational data assimilation system, ABC-DA. The purpose of ABC-DA is to permit quick and efficient research into data assimilation methods suitable
for convective scale systems. The system can also be used as an aid to teach and demonstrate data assimilation principles.

ABC-DA is flexible and configurable, and is efficient enough to be run on a personal computer. The system can run a number of assimilation methods (currently 3DVar and 3DFGAT have been implemented), with user configurable observation networks. Observation operators for direct observations and wind speeds are part of the current system, and these can be expanded relatively easily to include operators for Doppler winds for example. A key feature of any data assimilation system is how it
specifies the background error covariance matrix. ABC-DA uses a control variable transform method to allow this to be done efficiently. This version of ABC-DA mirrors many operational configurations, by modelling multivariate error covariances with uncorrelated control parameters, each with special uncorrelated spatial patterns.

The software developed does (amongst other things) model runs, calibration tasks associated with the background error covariance matrix, testing and diagnostic tasks, single data assimilation runs, multi-cycle assimilation/forecast experiments,
and has associated visualisation software.

As a demonstration, the system is used to tackle a scientific question concerning the role of geostrophic balance (GB) to model background error covariances between mass and wind fields. This question arises because, although GB is a very useful mechanism that is successfully exploited in larger scale assimilation systems, its use is questionable at convective scales due to the typically larger Rossby numbers where GB is not so relevant. A series of identical twin experiments is done in cycled
assimilation configurations. One experiment exploits GB to represent mass-wind covariances in a mirror of an operational set-up (with use of an additional vertical regression (VR) step, as used operationally). This experiment performs badly where error accumulates over time. Two further experiments are done: one that does not use GB, and another that does but without the VR step. Turning off GB impairs the performance, and turning off VR improves the performance in general. It is concluded that there is scope to further improve the way that the background error covariance matrices are represented at convective-scale.
Ideas for further possible developments of ABC-DA are discussed.

# 1 Introduction

The grid sizes of limited area models for operational weather forecasting have become small enough to allow some convective processes to be resolved explicitly (Clark et al., 2016; Yano et al., 2018). Some leading operational models include the COSMO (Consortium for Small scale MOdelling) model (Baldauf et al., 2011), used at MeteoSwiss (1.1 km grid size) and at DWD (2.8 km grid size); the AROME (Application of Research to Operations at Mesoscale) model (Brousseau et al., 2016), used at Météo-France (1.3 km grid size); the UKV (UK Variable resolution) model (Tang et al., 2013) (1.5 km grid size); and the WRF (US Weather Research and Forecasting) model (Schwartz and Liu, 2014) (3 km grid size). Each of these systems is invaluable in the forecasting of fine-scale weather, including that associated with convective storms, and has its own data assimilation (DA) system to estimate its initial conditions from new observations and a background state.

Apart from the capability to assimilate new high-resolution observation types, such as radar reflectivity and Doppler radial wind, the DA systems are still based on those designed for use with synoptic- and planetary-scale phenomena in mind. The convective-scale DA problem needs to account for effects that can often be safely ignored or treated approximately when dealing with large scales. These include certain dynamical properties of background state errors (namely non-hydrostatic and non-geostrophic contributions, vertical motion, multiple phases of water, strong inhomogeneity and flow-dependence, and non-Gaussianity), certain properties of observation errors (namely cross-correlations), and other features associated with a small grid size (e.g. feature misalignment). There are also challenges associated with assimilating new observation types (as mentioned above, including the large volumes of data needed), the short DA time window (often 1 hour or less), the compatibility of lateral boundary conditions from a coarser parent model, and questions concerning the appropriateness of allowing DA to simultaneously modify the larger scale flows present in the convective-scale problem.

The properties of background state errors is of particular concern to this paper, although the DA system to be described can be equally applied to study other aspects of convective-scale DA, such as the exploration of strategies for high-resolution observation networks (such as those from Doppler wind instruments), or indeed some of the advanced DA methods mentioned below. In DA, the background state is traditionally assumed to be subject to random error which is distributed according to a Gaussian distribution described by a multivariate error covariance matrix (the '$\mathbf{B}$-matrix', e.g. Bannister (2008a)). Given that $\mathbf{B}$ is too large to store explicitly, in variational DA (Var) it is represented in the form of a 'model' (Bannister, 2008b). One important means of representing $\mathbf{B}$ in a way that naturally adapts to the flow conditions is to derive a matrix implicitly from an ensemble of forecasts, which are often produced anyway for probabilistic forecasting purposes. This is the basis of the ensemble Kalman filter (e.g. Houtekamer and Zhang (2016)), and EnVar (pure ensemble-variational) formulations (e.g. Liu et al. (2008)). Although information from an ensemble in principle follows the dynamical properties of the model to be reflected in the $\mathbf{B}$-matrix used at the analysis time, the result is often corrupted by sampling error due to the small ensemble sizes (usually a few tens of members). For this reason, the $\mathbf{B}$-matrix used operationally is often still modelled according to physical insight. That insight though is based on traditional assumptions of (for instance) geophysical balance whose applicability are questionable at convective-scales.

Studying convective-scale DA in operational systems is burdened severely by the cost and complexity of these systems. The DA system described in this paper has been designed in the same spirit as that of the convective-scale 'toy' model (the 'ABC model', Petrie et al. (2017)), i.e. with an emphasis on low cost and simplicity. This DA system (together with the model code is hereafter called 'ABC-DA') is a multi-featured Var system suited to the ABC model. ABC-DA is actually a suite of software used not only to perform DA itself (in cycling mode if required), but also to calibrate the **B**-matrix from sets of forecast data, to flexibly generate randomly perturbed data such as synthetic observations from a 'truth' run (which can then be assimilated), to compute a sample of covariances implied from a chosen **B**-matrix model, and to perform a set of validation tests. The suite also includes sample plotting codes to help visualise and monitor the outputs, a script to build the executables, sample run scripts, and detailed user documentation. The ABC-DA numerical codes are written in Fortran-90, scripts are written in Linux Bash, and the plotting code is written in Python-2. Certain open-source software libraries are also required to compile the code.

This paper presents the scientific documentation for the system, with examples and pointers to how a user can access the software. Finally a short study of ABC-DA is presented to investigate the impact of balance constraints in the formulation of the convective-scale **B**-matrix. The paper is structured as follows. In Sect. 2 the ABC model is described, in Sect. 3 the ABC-DA system is outlined, in Sect. 4, the ABC-DA system is described in detail, in Section 5 a brief study of the role of geostrophic balance is presented, and in Sect. 6 the paper is summarised.

## 2 The 'ABC' model

### 2.1 The model equations

The ABC model comprises a set of simplified partial differential equations for a two-dimensional spatial grid ($x$ and $z$), plus time ($t$), which are based on the Euler equations. This section summarises the ABC model, and the reader is directed to Petrie et al. (2017) for the details. The model equations are as follows:

$$\frac{\partial u}{\partial t} + B\mathbf{u} \cdot \nabla u + C\frac{\partial \tilde{\rho}'}{\partial x} - fv = 0, \tag{1a}$$

$$\frac{\partial v}{\partial t} + B\mathbf{u} \cdot \nabla v + fu = 0, \tag{1b}$$

$$\frac{\partial w}{\partial t} + B\mathbf{u} \cdot \nabla w + C\frac{\partial \tilde{\rho}'}{\partial z} - b' = 0, \tag{1c}$$

$$\frac{\partial \tilde{\rho}'}{\partial t} + B\nabla \cdot (\tilde{\rho}\mathbf{u}) = 0, \tag{1d}$$

$$\frac{\partial b'}{\partial t} + B\mathbf{u} \cdot \nabla b' + A^2 w = 0, \tag{1e}$$

where $\mathbf{u} = \begin{pmatrix} u & v & w \end{pmatrix}$ is the wind vector (comprising zonal, meridional, and vertical wind components respectively), $\tilde{\rho}$ is the scaled density variable (akin to pressure), $b$ is the buoyancy variable (akin to temperature), $f$ is the Coriolis parameter, $g$ is the acceleration due to gravity, and $A$, $B$, and $C$ are tunable parameters (see below). Primed variables indicate perturbations from a reference state defined as $b(x,z,t) = g + b_0(z) + b'(x,z,t)$ and $\tilde{\rho}(x,z,t) = 1 + \tilde{\rho}'(x,z,t)$. The model supports a range of motions, namely balanced (Rossby-like) modes, and unbalanced (gravity and acoustic) modes, which have been studied in

detail in Petrie et al. (2017). There are three tunable parameters: $A$ is the pure gravity wave frequency, controlling the gravity wave speeds in the model; $B$ modulates the advective and divergent terms in the equations, controlling the acoustic wave speeds; and $C$ specifies the equation of state that relates pressure and density perturbations, $p' = C\rho_0\tilde{\rho}'$, where $\rho_0$ is a density scaling constant. In the linearised equations, only the product $BC$ (and not $B$ and $C$ individually) affects the characteristics of the flow, so $C$ also controls the acoustic wave speeds. In numerical integrations of the non-linear equations Eq. (1), the effect of scaling $C$ is found to be virtually indistinguishable from scaling $B$ in terms of the patterns of forecast perturbations.

## 2.2 Properties of the ABC model equations

Equations Eq. (1) conserve total mass and energy, although this exact property is lost when the equations are discretised for numerical integration. The equations approximate to geostrophic and hydrostatic balance (GB and HB respectively) when the Rossby number, $Ro = \mathcal{U}/f\mathcal{L}$ is small (where $\mathcal{U}$ is the characteristic zonal wind speed and $\mathcal{L}$ is the characteristic horizontal length-scale of the motion). The GB relations are

$$-fv + C\frac{\partial\tilde{\rho}'}{\partial x} \quad = \quad 0, \tag{2a}$$

$$u \quad = \quad 0, \tag{2b}$$

and the HB relation is

$$-b' + C\frac{\partial\tilde{\rho}'}{\partial z} = 0. \tag{3}$$

These balance relations are well satisfied for motion at the large scales where $Ro$ is small and are used in traditional Var schemes to model the covariances between mass, wind, and temperature perturbations in background errors. We will revisit these later in the paper.

## 2.3 Discretisation and integration

As reported in Petrie et al. (2017) the continuous equations Eq. (1) have been discretised in time and space (the current implementation uses a $360 \times 60$ (horizontal $\times$ vertical) element grid with a gridbox size of $1500 \times 250$ m. Variables are stored on an Arakawa C grid in the horizontal and Charney-Phillips in the vertical (see Fig. 1 of Petrie et al. (2017)), and periodic boundary conditions are imposed in the horizontal to avoid the need for a driving model to provide lateral boundary conditions. The integration scheme used is the split-explicit, forward-backward scheme of Cullen and Davies (1991) with a main timestep of $\Delta t = 4$ s.

## 2.4 Future developments of the model

The current version of the ABC model does not include moist processes. There is much that can be learned about convective-scale DA from a dry model, but assimilating and forecasting moisture fields is a major reason for convective-scale forecasting (Sun et al., 2014; Bannister et al., 2020). It is planned to upgrade the model to permit the advection of one or more water

variables and allow condensation and evaporation processes to affect the flow. The assimilation of moisture is a complex task and so a moist ABC-DA system is expected to be very useful in that line of research.

## 3   Overview of the ABC-DA system

Like the ABC model, the DA system is intended to be low cost and easy to run when compared to a operational-scale system, yet mirror many of the features and options available in operational systems. In this section we review the principles on which ABC-DA is based, which includes a definition of the mathematical notation, but we leave it to Sect. 4 to describe the details.

### 3.1   Variational data assimilation

Var systems construct a scalar functional (called a cost function, $J$) that is minimised with respect to the state $\mathbf{x}$ by considering observations made over a time window (indicated by an integer time index) $t \in [0, T]$:

$$J[\mathbf{x}] = \frac{1}{2}(\mathbf{x} - \mathbf{x}^{\mathrm{b}})\mathbf{B}^{-1}(\mathbf{x} - \mathbf{x}^{\mathrm{b}}) + \frac{1}{2}\sum_{t=0}^{T}[\mathbf{y}(t) - \mathbf{y}^{\mathrm{m}}(t)]^{\mathrm{T}}\mathbf{R}_t^{-1}[\mathbf{y}(t) - \mathbf{y}^{\mathrm{m}}(t)]. \tag{4}$$

In Eq. (4), $\mathbf{x}$ represents all variables of the model state at $t = 0$ (here $u$, $v$, $w$, $\tilde{\rho}'$, and $b'$) at each location in the domain, $\mathbf{x}^{\mathrm{b}}$ is a special state at $t = 0$ called the background state (normally a short forecast from the previous DA), $\mathbf{y}(t)$ is the collection

of observations at time $t$, and $\mathbf{y}^{\mathrm{m}}(t)$ is the model's version of the observations computed from $\mathbf{x}$. Let there be $n$ elements in $\mathbf{x}$ and $p_t$ elements in $\mathbf{y}(t)$. Here we use the convention that quantities like $\mathbf{x}$ without a time argument imply the value at $t = 0$. The model's observations are found in two steps: firstly the state is found at time $t$ using the model propagator: $\mathbf{x}(t) = \mathcal{M}_{0 \to t}(\mathbf{x})$, which is the result of integrating Eq. (1) from times 0 to $t$, and then the model's observations are found using the observation operator at this time: $\mathbf{y}^{\mathrm{m}} = \mathcal{H}_t(\mathbf{x}(t))$. This DA method is known as 4DVar (Dimet and Talagrand, 1986).

$\mathbf{B}$ and $\mathbf{R}_t$ are covariance matrices (e.g. Kalnay (2002)) pertaining to errors in the background state, and in the observations at time $t$ respectively. Mathematically $\mathbf{B}$ (an $n \times n$ matrix) and $\mathbf{R}_t$ (a $p_t \times p_t$ matrix) may be thought as the metrics in which deviations are measured in the cost function (i.e. measures of the precision to which the background and observations are known). The analysis, $\mathbf{x}^{\mathrm{a}}$ is the special state that minimises $J$.

The $\mathbf{B}$ and $\mathbf{R}_t$ matrices are important as they can have a profound effect on the way that observations combine with the

background to yield the analysis. It is a particular challenge to use a $\mathbf{B}$-matrix that is relevant to the uncertainties in convective-scale forecasts. As $\mathbf{B}$ is usually a much larger matrix than $\mathbf{R}_t$ (operationally by orders of magnitude), it cannot practically be stored explicitly. Instead the $\mathbf{B}$-matrix is modelled, which is usually done via the technique of control variable transforms (CVTs) – see Sect. 3.4. As this modelling process is a major component of any DA system, and may require new thinking for convective-scale systems, much of the design of ABC-DA is concerned with how $\mathbf{B}$ is modelled. As a starting point for

ABC-DA, the approach that is currently implemented is a conventional one (to mirror typical current operational configurations that were designed around global systems, Sect. 4.2, but still applied for convective-scale systems). This approach though can be adapted to accommodate new convective-scale strategies that are discussed at the end of the paper.

## 3.2 The incremental formulation of the problem

If $\mathcal{M}_{0 \to t}$ and $\mathcal{H}_t$ are linear functions, then Eq. (4) is a quadratic function of $\mathbf{x}$ and may be minimised using efficient algorithms such as a conjugate gradient-based method (Golub and Van Loan, 1996; Lewis et al., 2006). The model $\mathcal{M}_{0 \to t}$ though is a non-linear operator, and many observations require non-linear observation operators (such as measurements of wind speed, top-of-atmosphere radiance, etc.). This leads to a non-quadratic function, which may have multiple minima. Furthermore there are no general efficient minimising algorithms for such problems. In order to simplify the problem, the cost function is minimised by breaking it down into a sequence of quadratic problems by iteratively linearising $\mathcal{M}_{0 \to t}$ and $\mathcal{H}_t$. This is incremental Var (Courtier et al., 1994).

Suppose that $\mathbf{x}^{\mathrm{r}}(t)$ is a reference trajectory satisfying $\mathbf{x}^{\mathrm{r}}(t) = \mathcal{M}_{t-1 \to t}(\mathbf{x}^{\mathrm{r}}(t-1))$, $1 \le t \le T$. A perturbation to this trajectory ($\delta$ prefix) at $t$ is approximately related to a perturbation at $t-1$ via the linear operation $\delta \mathbf{x}(t) \approx \mathbf{M}_{t-1 \to t} \delta \mathbf{x}(t-1)$, where the full states are $\mathbf{x}(t) \approx \mathbf{x}^{\mathrm{r}}(t) + \delta \mathbf{x}(t)$ and $\mathbf{x}(t-1) \approx \mathbf{x}^{\mathrm{r}}(t-1) + \delta \mathbf{x}(t-1)$. $\mathbf{M}_{t-1 \to t}$ is the tangent linear (or Jacobian) of $\mathcal{M}_{t-1 \to t}$ and is mathematically representable by an $n \times n$ matrix. We assume that $\mathbf{x}^{\mathrm{r}}(t) + \delta \mathbf{x}(t)$ is close to $\mathcal{M}_{t-1 \to t}(\mathbf{x}^{\mathrm{r}}(t-1) + \delta \mathbf{x}(t-1))$ providing $\delta \mathbf{x}(t-1)$ is sufficiently small. Similarly, suppose that the observation values computed from $\mathbf{x}^{\mathrm{r}}(t)$ satisfy $\mathbf{y}^{\mathrm{mr}}(t) = \mathcal{H}_t(\mathbf{x}^{\mathrm{r}}(t))$. A perturbation to these reference observations is approximately related to a perturbation in $\mathbf{x}^{\mathrm{r}}(t)$ via the linear operation $\delta \mathbf{y}(t) = \mathbf{H}_t \delta \mathbf{x}(t)$. $\mathbf{H}_t$ is the tangent linear (or Jacobian) of $\mathcal{H}_t$ and is mathematically representable by a $p_t \times n$ matrix. These approximations may be summarised as the following:

$$\mathbf{x} \approx \mathbf{x}^{\mathrm{r}} + \delta \mathbf{x}, \tag{5a}$$

$$\mathbf{y}^{\mathrm{m}}(t) \approx \mathbf{y}^{\mathrm{mr}}(t) + \delta \mathbf{y}^{\mathrm{m}}(t), \tag{5b}$$

$$\text{where } \delta \mathbf{y}^{\mathrm{m}}(t) = \mathbf{H}_t \mathbf{M}_{0 \to t} \delta \mathbf{x}. \tag{5c}$$

When Eq. (5) and the following definitions:

$$\delta \mathbf{x}^{\mathrm{b}} = \mathbf{x}^{\mathrm{b}} - \mathbf{x}^{\mathrm{r}}, \tag{6a}$$

$$\mathbf{d}(t) = \mathbf{y}(t) - \mathcal{H}_t(\mathcal{M}_{0 \to t}(\mathbf{x}^{\mathrm{r}})) = \mathbf{y}(t) - \mathbf{y}^{\mathrm{mr}}(t), \tag{6b}$$

are substituted into Eq. (4), $J$ becomes a functional of the perturbation $\delta \mathbf{x}$ instead of the full state $\mathbf{x}$:

$$J[\delta \mathbf{x}] = \frac{1}{2} \left( \delta \mathbf{x} - \delta \mathbf{x}^{\mathrm{b}} \right)^{\mathrm{T}} \mathbf{B}^{-1} \left( \delta \mathbf{x} - \delta \mathbf{x}^{\mathrm{b}} \right) + \frac{1}{2} \sum_{t=0}^{T} [\mathbf{H}_t \mathbf{M}_{0 \to t} \delta \mathbf{x} - \mathbf{d}(t)]^{\mathrm{T}} \mathbf{R}_t^{-1} [\mathbf{H}_t \mathbf{M}_{0 \to t} \delta \mathbf{x} - \mathbf{d}(t)]. \tag{7}$$

This is the incremental form of 4DVar. $J[\delta \mathbf{x}]$ is exactly quadratic, allowing efficient algorithms to be used to minimise it to yield the special state $\delta \mathbf{x}^{\mathrm{a}}$. The iterations required to minimise Eq. (7) form an *inner loop*. The full cost function Eq. (4) is minimised by updating the reference state, $\mathbf{x}^{\mathrm{r}} \to \mathbf{x}^{\mathrm{r}} + \delta \mathbf{x}^{\mathrm{a}}$, and repeating the inner loops. These iterations form the *outer loop*. In the first outer loop, $\mathbf{x}^{\mathrm{r}}$ is typically set to $\mathbf{x}^{\mathrm{b}}$. This inner/outer loop procedure though does not necessarily find the global minimum of Eq. (4), which can lead to complications in highly non-linear systems with a long DA time window (e.g. Fabry and Sun (2010)).

Because $\mathbf{M}_{0\rightarrow t}$ is a difficult operator to derive, the approximation that $\mathbf{M}_{0\rightarrow t} = \mathbf{I}$ is often made. This leads to an approximate method called 3DFGAT (3DVar First Guess at Appropriate Time, Lee et al. (2004); Lawless (2010)). For applications when it is too expensive to use $\mathcal{M}_{0\rightarrow t}$ in the DA loops, a further approximation is made that $\mathbf{x}(t) = \mathbf{x}$ over the window. This is called 3DVar, although many system that specify 3DVar actually use 3DFGAT.

### 3.3 The observations, their operators, and their error statistics

The observation operator $\mathcal{H}_t$ (and $\mathbf{H}_t$) are built to suit the range observations assimilated. The components of $\mathcal{H}_t$ and $\mathbf{H}_t$ represent the model's version of the observations. Typical examples include simple bi-linear interpolation of grid values to an observation location, the computation of model wind speed as the root of the sum of squares of the wind components, or the evaluation of top-of-atmosphere radiance by a radiative transfer equation. The ABC-DA system currently implements observations of the first two kinds, but the system is flexible enough to support observations of any required model variable at arbitrary times and positions. The $\mathbf{R}_t$-matrices are taken to be diagonal, and have specified variances. The system could be adapted to extend any of these aspects to include more complicated observation operators, such as radiative transfer models, or Doppler winds, or for correlated observation errors.

### 3.4 Modelling B with control variable transforms

The $\mathbf{B}$-matrix is meant to represent the covariances of errors in $\mathbf{x}^{\mathrm{b}}$. Operational-scale DA systems all share the challenge of determining and using $\mathbf{B}$ given that this $n \times n$ matrix is too large to manipulate (or even store) and is in any case unknowable. Most practical variational methods use *control variable transforms* (CVTs) to simplify this problem. Consider a vector $\delta\boldsymbol{\chi}$, which is an alternative representation of $\delta\mathbf{x}$ via the relation

$$\delta\mathbf{x} = \mathbf{U}\delta\boldsymbol{\chi}. \tag{8}$$

$\delta\boldsymbol{\chi}$ is called a *control vector*, and $\mathbf{U}$ is the CVT. The CVT is a powerful way of accounting for cross-correlations between background errors of model variables (including spatial and multivariate components). $\delta\mathbf{x}$ and $\delta\boldsymbol{\chi}$ have different assumed statistical properties: model space errors have covariance $\left\langle \delta\mathbf{x}\delta\mathbf{x}^{\mathrm{T}} \right\rangle_{\mathrm{b}} = \mathbf{B}$, and control variables are taken to be uncorrelated and have unit variance $\left\langle \delta\boldsymbol{\chi}\delta\boldsymbol{\chi}^{\mathrm{T}} \right\rangle_{\mathrm{b}} = \mathbf{I}$, where the 'b' subscript indicates expectation over hypothetical background error samples. When Eq. (8) is substituted into Eq. (7) $J$ becomes a functional of $\delta\boldsymbol{\chi}$:

$$J[\delta\boldsymbol{\chi}] = \frac{1}{2}\left(\delta\boldsymbol{\chi} - \delta\boldsymbol{\chi}^{\mathrm{b}}\right)^{\mathrm{T}}\left(\delta\boldsymbol{\chi} - \delta\boldsymbol{\chi}^{\mathrm{b}}\right) + \frac{1}{2}\sum_{t=0}^{T}\left[\mathbf{H}_t\mathbf{M}_{0\rightarrow t}\mathbf{U}\delta\boldsymbol{\chi} - \mathbf{d}(t)\right]^{\mathrm{T}}\mathbf{R}_t^{-1}\left[\mathbf{H}_t\mathbf{M}_{0\rightarrow t}\mathbf{U}\delta\boldsymbol{\chi} - \mathbf{d}(t)\right]. \tag{9}$$

Notice that the $\mathbf{B}$-matrix has effectively disappeared from this formulation, because of the assumed statistical properties of $\delta\boldsymbol{\chi}$. This is a considerable simplification, although the problem of defining the CVT remains. The $\delta\boldsymbol{\chi}$ cost function is minimised (giving the special vector $\delta\boldsymbol{\chi}^{\mathrm{a}}$), which then leads to $\delta\mathbf{x}^{\mathrm{a}}$ via Eq. (8). This is equivalent to minimising Eq. (7) with a background error covariance matrix equal to

$$\mathbf{B}^{\mathrm{ic}} = \mathbf{U}\mathbf{U}^{\mathrm{T}}, \tag{10}$$

which is known as the *implied background error covariance matrix*. Apart from not needing to know the $\mathbf{B}$-matrix explicitly, the minimisation problem Eq. (9) is found to be numerically better conditioned than Eq. (7), leading to a more efficient and accurate minimisation.

It is common to work backwards here: first a CVT is proposed (based on physical principles such as those discussed in Bannister (2008b) and later in this paper), and then its ability to generate reasonable background error covariance structures is studied by looking at the implied covariances. This can be done by studying either $\mathbf{U}\mathbf{U}^{\mathrm{T}}$, or the analysis increments of single-observation DA experiments. Constructing $\mathbf{U}$ is one way of doing *background error covariance modelling*. We do this by defining new parameters and their spatial covariances via a proposed form like $\mathbf{U} = \mathbf{U}_{\mathrm{p}}\mathbf{U}_{\mathrm{s}}$ (Bannister, 2008b). Here $\mathbf{U}_{\mathrm{p}}$ is the *parameter transform* (where $\mathbf{U}_{\mathrm{p}}^{-1}$ transforms model variables to alternative parameters that are assumed uncorrelated using sets of balance operators as in Parrish and Derber (1992); Gauthier et al. (1999)), and $\mathbf{U}_{\mathrm{s}}$ is the *spatial transform* (which transforms each parameter's field to modes that are assumed to be uncorrelated, such as Fourier modes). $\mathbf{U}_{\mathrm{s}}$ can itself be decomposed into separate horizontal, vertical parts, and scaling parts, e.g. $\mathbf{U}_{\mathrm{s}} = \mathbf{\Sigma}\mathbf{U}_{\mathrm{v}}\mathbf{U}_{\mathrm{h}}$ (see Sect. 4.2.3). More complicated sequences of transforms are also possible, e.g. based on wavelets (Deckmyn and Berre, 2005). A property of the CVT approach is that $\mathbf{B}$ can be modelled even if it is singular.

## 3.5 The gradient of $J$ and minimising the cost function

Equation Eq. (7) is minimised by iteratively adjusting $\delta\boldsymbol{\chi}$ until a convergence criterion is met, indicating that a point close to the minimum of $J[\delta\boldsymbol{\chi}]$ has been found. The gradient vector, $\nabla_{\delta\boldsymbol{\chi}}J$, is used with a conjugate gradient algorithm to perform this task. $\nabla_{\delta\boldsymbol{\chi}}J$ is found by differentiating $J[\delta\boldsymbol{\chi}]$:

$$\nabla_{\delta\boldsymbol{\chi}}J = \delta\boldsymbol{\chi} - \delta\boldsymbol{\chi}^{\mathrm{b}} + \mathbf{U}^{\mathrm{T}}\sum_{t=0}^{T}\mathbf{M}_{0\to t}^{\mathrm{T}}\mathbf{H}_t^{\mathrm{T}}\mathbf{R}_t^{-1}\left[\mathbf{H}_t\mathbf{M}_{0\to t}\mathbf{U}\delta\boldsymbol{\chi} - \mathbf{d}(t)\right], \tag{11}$$

where $\mathbf{d}(t)$ is the difference between the observations at time $t$ and the model's version of them based on the reference state Eq. (6b). Equation Eq. (11) requires the Jacobians $\mathbf{H}_t$ and $\mathbf{M}_{0\to t}$, the CVT $\mathbf{U}$, and their adjoint counterparts. The evaluation of Eq. (11) can be made more efficient by the following standard algorithm.

1. Set the reference state at $t = 0$ to the background state, $\mathbf{x}^{\mathrm{r}} = \mathbf{x}^{\mathrm{b}}$.

2. Do the outer loop.

   (a) For the first outer loop, $\delta\boldsymbol{\chi}^{\mathrm{b}} = 0$, otherwise compute $\delta\boldsymbol{\chi}^{\mathrm{b}} = \mathbf{U}^{-1}\left(\mathbf{x}^{\mathrm{b}} - \mathbf{x}^{\mathrm{r}}\right)$.

   (b) Compute $\mathbf{x}^{\mathrm{r}}(t)$ over the time window, $1 \le t \le T$ with the non-linear model, $\mathbf{x}^{\mathrm{r}}(t) = \mathcal{M}_{t-1\to t}\left(\mathbf{x}^{\mathrm{r}}(t-1)\right)$.

   (c) Compute the reference state's observations, $\mathbf{y}^{\mathrm{mr}}(t) = \mathcal{H}_t\left(\mathbf{x}^{\mathrm{r}}(t)\right)$.

   (d) Compute the differences, $\mathbf{d}(t) = \mathbf{y}(t) - \mathbf{y}^{\mathrm{mr}}(t)$.

   (e) Set $\delta\boldsymbol{\chi} = 0$ and $\delta\mathbf{x} = 0$.

   (f) Do the inner loop.

i. Integrate the perturbation trajectory over the time window, $1 \leq t \leq T$, with the linear forecast model, $\delta\mathbf{x}(t) = \mathbf{M}_{t-1\to t}\delta\mathbf{x}(t-1)$.

ii. Compute the perturbations to the model observations, $\delta\mathbf{y}^{\mathrm{m}}(t) = \mathbf{H}_t\delta\mathbf{x}(t)$.

iii. Compute $\boldsymbol{\Delta}(t)$ vectors defined as $\boldsymbol{\Delta}(t) = \mathbf{H}_t^{\mathrm{T}}\mathbf{R}_t^{-1}\left[\delta\mathbf{y}^{\mathrm{m}}(t) - \mathbf{d}(t)\right]$.

iv. Set the adjoint state $\boldsymbol{\lambda}(T+1) = 0$.

v. Integrate the following adjoint equation backwards in time, $T \geq t \geq 0$, $\boldsymbol{\lambda}(t) = \boldsymbol{\Delta}(t) + \mathbf{M}_{t\to t+1}^{\mathrm{T}}\boldsymbol{\lambda}(t+1)$.

vi. Compute the gradient as follows, $\nabla_{\delta\boldsymbol{\chi}}J = \delta\boldsymbol{\chi} - \delta\boldsymbol{\chi}^{\mathrm{b}} + \mathbf{U}^{\mathrm{T}}\boldsymbol{\lambda}(0)$.

vii. Use the conjugate gradient algorithm to adjust $\delta\boldsymbol{\chi}$ to reduce the value of $J$.

viii. Compute the new increment in model space using the CVT, $\delta\mathbf{x} = \mathbf{U}\delta\boldsymbol{\chi}$.

ix. Go to step 2(f)i until inner loop convergence criterion is satisfied.

(g) Update the reference state, $\mathbf{x}^{\mathrm{r}} \to \mathbf{x}^{\mathrm{r}} + \delta\mathbf{x}$.

(h) Go to step 2a until outer loop convergence criterion is satisfied. At convergence set $\mathbf{x}^{\mathrm{a}} = \mathbf{x}^{\mathrm{r}}$.

3. Run a non-linear forecast from $\mathbf{x}^{\mathrm{a}}$ for the background of the next cycle, and longer forecasts if required.

The full procedure (adapted for 3DFGAT, where the linear model is omitted) is shown graphically in Sect. 4.7.

## 3.6 System tests

The system has a special suite to check aspects of operators that are coded. Operators that have an adjoint counterpart are subject to an adjoint test to demonstrate that the adjoint has been coded correctly. This includes the linearised observation operators, and components of $\mathbf{U}$. Many of these operators are sub-divided into constituents that are tested separately (e.g. interpolation, halo swapping, and Fourier transforms). For a coded operator, $\mathbf{A}$, with input $\mathbf{v}_{\mathrm{in}}$, and its coded adjoint $\mathbf{A}^{\mathrm{T}}$, the adjoint test computes the left- and right-hand sides of the following formula, which must agree to machine precision to gain confidence that the coded adjoint is correct:

$$\left(\mathbf{A}\mathbf{v}_{\mathrm{in}}\right)^{\mathrm{T}}\mathbf{A}\mathbf{v}_{\mathrm{in}} \overset{?}{=} \mathbf{v}_{\mathrm{in}}^{\mathrm{T}}\mathbf{A}^{\mathrm{T}}\mathbf{A}\mathbf{v}_{\mathrm{in}}, \tag{12}$$

where a random vector $\mathbf{v}_{\mathrm{in}}$ will normally suffice. The CVT needs to be inverted in the gradient algorithm when using more than one outer loop (step 2a in Sect. 3.5), and in calibrating the $\mathbf{B}$-matrix (Sect. 4.3). These operators are subject to an *inverse test* to demonstrate that the inverse has been coded correctly. This is done by reading in a perturbation state, and then passing it through $\mathbf{A}\mathbf{A}^{-1}$. The result is output, which can be compared to the original field read-in. A test that the gradient of the cost function (as computed for the minimisation) is valid can be confirmed in a gradient test, which is also provided as part of the test suite. The gradient test estimates progressively more accurate finite-difference approximations to the gradient and checks that they converge to the analytically computed gradient Eq. (11). Other tests are possible that have not been included in this version of ABC-DA, e.g. checks that the innovation statistics, namely $\left\langle (\mathbf{y} - \mathbf{y}^{\mathrm{m}})(\mathbf{y} - \mathbf{y}^{\mathrm{m}})^{\mathrm{T}} \right\rangle$ equals to $\mathbf{R} + \mathbf{H}\mathbf{B}^{\mathrm{ic}}\mathbf{H}^{\mathrm{T}}$ (where $\mathbf{y}^{\mathrm{m}}(t)$ is the background's version of the observations and the angled brackets indicate average over a large number of DA cycles with the same observation network).

## 4 Scientific and technical configuration of ABC-DA v1.4

This section is a description of the current scientific configuration of the ABC-DA system. This section also contains some technical information and can be read in conjunction with the user documentation available on GitHub at (*ABC-DA_1.4da/docs/Documentation.pdf*), where more information is available, including names of the executables to be run, the namelist variables that have to be set, and the input and output file names. References are made to this document in the sections below in the form of *GitHubDoc§x*.

The code is divided into master programs that perform specific tasks. The relevant variables are set in a namelist file (filenames, options, switches, and parameters), and then the relevant executable is run. A list of the available master routines is listed in *GitHubDoc§2*, and instructions on how to download and build the code are found in *GitHubDoc§3*.

### 4.1 Construction of a model state and making a forecast

The initial conditions for a model run may be generated using the program *Master_PrepareABC_InitState* (*GitHubDoc§4.1*). The code can take a slice from a specific Met Office Unified Model (UM) file, or can generate a simple idealised pressure 'blob' of specified position and size (or a combination of these). When initial fields are taken from the UM, they need to be adjusted to make them compatible with the ABC model. This involves a number of steps: (i) adjusting the fields towards the E and W edges of the domain to be consistent with the periodic boundary conditions; (ii) adding a constant to the $v$ field to force its integral over each level to zero to allow GB Eq. (2a) (balance condition Eq. (2b) though is not enforced to allow some imbalance); (iii) computing $\tilde{\rho}'$ from $v$ with Eq. (2a), and then adding a constant to force its integral over each level to zero; (iv) computing $b'$ to satisfy the HB condition Eq. (3); and (v) setting $w'$ so that the 3D winds have zero divergence. A forecast can be made from these initial conditions using the program *Master_RunNLModel* (*GitHubDoc§4.2*) by numerically integrating equations Eq. (1).

### 4.2 The CVTs (B-matrix) implemented

ABC-DA has a variety of options implemented to model the **B**-matrix using control variable transforms (CVTs, Sect. 3.4) and this section describes the current implementation. The transforms are most easily understood by describing first the inverse CVTs (since they allow the difference 'spaces' to be defined starting in model space and working towards the control space). The CVT operators defined are used in many of the programs mentioned in later sections.

### 4.2.1 The inverse parameter transform, $\mathbf{U}_\mathrm{p}^{-1}$

Recall that $\mathbf{U}_\mathrm{p}^{-1}$ transforms a perturbation in model variables Eq. (1) to alternative parameters that are assumed to be uncorrelated. It is needed primarily to calibrate the CVT (Sect. 4.3). The input fields in this procedure are the perturbations: $\delta\mathbf{u}$, $\delta\mathbf{v}$, $\delta\mathbf{w}$, $\delta\tilde{\rho}'$; and $\delta\mathbf{b}'$, and the output fields are (in the version of the code documented): $\delta\psi$ (stream-function), $\delta\boldsymbol{\chi}_\mathrm{vp}$ (velocity potential[1]), $\delta\tilde{\rho}'^\mathrm{u}$ (unbalanced scaled density), $\delta\mathbf{b}'^\mathrm{u}$ (unbalanced buoyancy), and $\delta\mathbf{w}^\mathrm{u}$ (unbalanced vertical wind). All input and output fields are a function of longitude and height. This is the algorithm for $\mathbf{U}_\mathrm{p}^{-1}$.

---

[1]Do not confuse the velocity potential perturbation, $\delta\boldsymbol{\chi}_\mathrm{vp}$, with the control vector, $\delta\boldsymbol{\chi}$.

1. Compute the stream-function:

$$\delta\boldsymbol{\psi} = \nabla_x^{-1}\delta\mathbf{v}. \tag{13}$$

The operator $\nabla_x^{-1}$ is defined as $\nabla_x^{-1}\delta\mathbf{v} = (\partial/\partial x)^{-2}\partial(\delta v)/\partial x$, which is based on application of the Helmholtz theorem (see e.g. Petrie et al. (2017), Sect. 4.1).

2. Compute the velocity potential (again based on the Helmholtz theorem):

$$\delta\boldsymbol{\chi}_{\mathrm{vp}} = \nabla_x^{-1}\delta\mathbf{u}. \tag{14}$$

Note that in Eq. (13) $\delta\psi$ depends only on the meridional wind and in Eq. (14) $\delta\boldsymbol{\chi}_{\mathrm{vp}}$ depends only on the zonal wind. This is unlike a system that has latitude dependence where $\delta\psi$ and $\delta\boldsymbol{\chi}_{\mathrm{vp}}$ would each depend on both $\delta\mathbf{u}$ and $\delta\mathbf{v}$ as per the Helmholtz theorem.

3. Compute the GB scaled density:

$$\delta\tilde{\boldsymbol{\rho}}'^{\mathrm{b}} = \alpha f\delta\psi/C, \tag{15}$$

which follows from application of the Helmholtz theorem for this system, $(\delta u, \delta v) = (\partial\delta\chi_{\mathrm{vp}}/\partial x, \partial\delta\psi/\partial x)$, applied to the GB equation Eq. (2a). The value $\alpha = 1$, unless the system is configured to turn off GB in this transform, in which case $\alpha = 0$.

4. Compute the balanced scaled density after it has been vertically regressed:

$$\delta\tilde{\boldsymbol{\rho}}'^{\mathrm{br}} = \mathbf{R}_\rho\delta\tilde{\boldsymbol{\rho}}'^{\mathrm{b}}. \tag{16}$$

The vertical regression (VR) operator $\mathbf{R}_\rho$ has the form $\mathbf{R}_\rho = \mathbf{C}^{\delta\tilde{\rho}'\delta\tilde{\rho}'^{\mathrm{b}}}\left(\mathbf{C}^{\delta\tilde{\rho}'^{\mathrm{b}}\delta\tilde{\rho}'^{\mathrm{b}}}\right)^{-1}$, where $\mathbf{C}^{\delta\tilde{\rho}'^{\mathrm{b}}\delta\tilde{\rho}'^{\mathrm{b}}}$ is the correlation matrix between a previously computed population of $\delta\tilde{\rho}'^{\mathrm{b}}$ perturbations with itself, and $\mathbf{C}^{\delta\tilde{\rho}'\delta\tilde{\rho}'^{\mathrm{b}}}$ is the correlation matrix between $\delta\tilde{\rho}'^{\mathrm{b}}$ and $\delta\tilde{\rho}'$. The justification for the use of $\mathbf{R}_\rho$ is given in Appendix B. The system can be configured to turn off this step (and is not used anyway if $\alpha = 0$ in in step 3).

5. Compute the unbalanced scaled density:

$$\delta\tilde{\boldsymbol{\rho}}'^{\mathrm{u}} = \delta\tilde{\boldsymbol{\rho}}' - \delta\tilde{\boldsymbol{\rho}}'^{\mathrm{br}} \tag{17}$$

($\delta\tilde{\boldsymbol{\rho}}'^{\mathrm{u}} = \delta\tilde{\boldsymbol{\rho}}'$ if $\alpha = 0$).

6. Compute the HB buoyancy:

$$\delta\mathbf{b}'^{\mathrm{b}} = \beta\mathbf{L}^{\mathrm{hb}}\delta\tilde{\boldsymbol{\rho}}'. \tag{18}$$

The operator $\mathbf{L}^{\mathrm{hb}}$ is defined as $\mathbf{L}^{\mathrm{hb}}\delta\tilde{\rho}' = C\partial\tilde{\rho}'/\partial z$, as in Eq. (3). The value $\beta = 1$, unless the system is configured to turn off HB, in which case $\beta = 0$.

7. Compute the unbalanced buoyancy:

$$\delta \mathbf{b}'^{\mathrm{u}} = \delta \mathbf{b}' - \delta \mathbf{b}'^{\mathrm{b}} \tag{19}$$

($\delta \mathbf{b}'^{\mathrm{u}} = \delta \mathbf{b}'$ if $\beta = 0$).

8. Compute the anelastically balanced vertical wind:

$$\delta \mathbf{w}^{\mathrm{b}} = \gamma \left( \mathbf{L}_u^{\mathrm{ab}} \delta \mathbf{u} + \mathbf{L}_{\tilde{\rho}'}^{\mathrm{ab}} \delta \tilde{\boldsymbol{\rho}}' \right). \tag{20}$$

Using Eq. (1d), the operators $\mathbf{L}_u^{\mathrm{ab}}$ and $\mathbf{L}_{\tilde{\rho}'}^{\mathrm{ab}}$ are defined as $\mathbf{L}_u^{\mathrm{ab}} \delta \mathbf{u} = -(1/\tilde{\rho}_0) \int dz'\, \partial(\tilde{\rho}_0 \delta u)/\partial x$ and $\mathbf{L}_{\tilde{\rho}'}^{\mathrm{ab}} \delta \tilde{\boldsymbol{\rho}}' = -(1/\tilde{\rho}_0) \int dz'\, [\partial(u_0 \delta \tilde{\rho}')/\partial x + \partial(w_0 \delta \tilde{\rho}')/\partial z']$ (integrating from the ground to height $z$), and $\delta \mathbf{w}^{\mathrm{b}}$ is the component of the vertical wind that, with $\delta \mathbf{u}$, has zero 3D divergence (sometimes called *anelastic balance* (AB), see Sect. 3.1.1 of Pielke (2002)). The value $\gamma = 1$, unless the system is configured to turn off AB, in which case $\gamma = 0$.

9. Compute the unbalanced vertical wind:

$$\delta \mathbf{w}^{\mathrm{u}} = \delta \mathbf{w} - \delta \mathbf{w}^{\mathrm{b}} \tag{21}$$

($\delta \mathbf{w}^{\mathrm{u}} = \delta \mathbf{w}$ if $\gamma = 0$).

Some of these steps may be omitted according to user options, as specified above. The above steps may be written more compactly as the following 'super matrix':

$$\delta \boldsymbol{\chi} = \mathbf{U}_{\mathrm{p}}^{-1} \delta \mathbf{x},$$

$$
\begin{pmatrix}
\delta \boldsymbol{\psi} \\
\delta \boldsymbol{\chi}_{\mathrm{vp}} \\
\delta \tilde{\boldsymbol{\rho}}'^{\mathrm{u}} \\
\delta \mathbf{b}'^{\mathrm{u}} \\
\delta \mathbf{w}^{\mathrm{u}}
\end{pmatrix}
=
\begin{pmatrix}
\mathbf{0} & \nabla_x^{-1} & \mathbf{0} & \mathbf{0} & \mathbf{0} \\
\nabla_x^{-1} & \mathbf{0} & \mathbf{0} & \mathbf{0} & \mathbf{0} \\
\mathbf{0} & -\alpha \mathbf{R}_\rho \frac{f}{C} \nabla_x^{-1} & \mathbf{0} & \mathbf{1} & \mathbf{0} \\
\mathbf{0} & \mathbf{0} & \mathbf{0} & -\beta \mathbf{L}^{\mathrm{hb}} & \mathbf{1} \\
-\gamma \mathbf{L}_u^{\mathrm{ab}} & \mathbf{0} & \mathbf{1} & -\gamma \mathbf{L}_{\tilde{\rho}'}^{\mathrm{ab}} & \mathbf{0}
\end{pmatrix}
\begin{pmatrix}
\delta \mathbf{u} \\
\delta \mathbf{v} \\
\delta \mathbf{w} \\
\delta \tilde{\boldsymbol{\rho}}' \\
\delta \mathbf{b}'
\end{pmatrix}. \tag{22}
$$

It is noted here that this particular form of transform is not *necessarily* the most appropriate form for convective-scale systems, e.g. GB in step 3 and HB in step 6 *may* not be relevant. There is, however, expected to be some GB at the larger scales represented, and HB at even shorter scales. Furthermore these relationships are still used in some operational systems, so their inclusion in this study is justified. The use of other balance relationships is possible, including statistical balance relationships (e.g. Derber and Bouttier (1999); Chen et al. (2013); Bannister et al. (2020)). An alternative balance relationship that may be applicable at convective-scale is mentioned in the summary.

### 4.2.2 The forward parameter transform, $\mathbf{U}_{\mathrm{p}}$

$\mathbf{U}_{\mathrm{p}}$ transforms perturbations of parameters to model space. This transform (and its adjoint) is used at each iteration of the Var algorithm. The input fields in this procedure are the parameter field perturbations: $\delta \boldsymbol{\psi}$, $\delta \boldsymbol{\chi}_{\mathrm{vp}}$, $\delta \tilde{\boldsymbol{\rho}}'^{\mathrm{u}}$, $\delta \mathbf{b}'^{\mathrm{u}}$, and $\delta \mathbf{w}^{\mathrm{u}}$; and the output fields are $\delta \mathbf{u}$, $\delta \mathbf{v}$, $\delta \mathbf{w}$, $\delta \tilde{\boldsymbol{\rho}}'$, and $\delta \mathbf{b}'$. This is the algorithm for $\mathbf{U}_{\mathrm{p}}$.

1. Compute the zonal wind based on the Helmholtz theorem:

$$\delta \mathbf{u} = \nabla_x \delta \boldsymbol{\chi}_{\mathrm{vp}}. \tag{23}$$

2. Compute the meridional wind (also based on the Helmholtz theorem):

$$\delta \mathbf{v} = \nabla_x \delta \boldsymbol{\psi}. \tag{24}$$

3. Compute the balanced scaled density, $\tilde{\rho}'^{\mathrm{b}}$ Eq. (15).

4. Compute the vertically regressed balanced scaled density, $\delta\tilde{\rho}'^{\mathrm{br}}$ Eq. (16).

5. Compute the total scaled density:

$$\delta\tilde{\boldsymbol{\rho}}' = \delta\tilde{\boldsymbol{\rho}}'^{\mathrm{br}} + \delta\tilde{\boldsymbol{\rho}}'^{\mathrm{u}}. \tag{25}$$

6. Compute the hydrostatically balanced buoyancy, $\delta\mathbf{b}'^{\mathrm{b}}$ Eq. (18).

7. Compute the total buoyancy:

$$\delta\mathbf{b}' = \delta\mathbf{b}'^{\mathrm{b}} + \delta\mathbf{b}'^{\mathrm{u}}.$$

8. Compute the anelastically balanced vertical wind, $\delta\mathbf{w}^{\mathrm{b}}$ Eq. (20).

9. Compute the total vertical wind:

$$\delta\mathbf{w} = \delta\mathbf{w}^{\mathrm{b}} + \delta\mathbf{w}^{\mathrm{u}}. \tag{26}$$

Again, some of these steps may be omitted according to user options, as set out in Sect. (4.2.1). The above steps may be written more compactly as the super matrix:

$$
\begin{aligned}
\delta\mathbf{x} &= \mathbf{U}_{\mathrm{p}}\delta\boldsymbol{\chi}, \\
\begin{pmatrix} \delta\mathbf{u} \\ \delta\mathbf{v} \\ \delta\mathbf{w} \\ \delta\tilde{\boldsymbol{\rho}}' \\ \delta\mathbf{b}' \end{pmatrix} &=
\begin{pmatrix}
\mathbf{0} & \nabla_x & \mathbf{0} & \mathbf{0} & \mathbf{0} \\
\nabla_x & \mathbf{0} & \mathbf{0} & \mathbf{0} & \mathbf{0} \\
\alpha\gamma\mathbf{L}_{\tilde{\rho}'}^{\mathrm{ab}}\mathbf{R}_\rho \frac{f}{C} & \gamma\mathbf{L}_u^{\mathrm{ab}}\nabla_x & \gamma\mathbf{L}_{\tilde{\rho}'}^{\mathrm{ab}} & \mathbf{0} & \mathbf{1} \\
\alpha\mathbf{R}_\rho \frac{f}{C} & \mathbf{0} & \mathbf{1} & \mathbf{0} & \mathbf{0} \\
\alpha\beta\mathbf{L}^{\mathrm{hb}}\mathbf{R}_\rho \frac{f}{C} & \mathbf{0} & \beta\mathbf{L}^{\mathrm{hb}} & \mathbf{1} & \mathbf{0}
\end{pmatrix}
\begin{pmatrix} \delta\boldsymbol{\psi} \\ \delta\boldsymbol{\chi}_{\mathrm{vp}} \\ \delta\tilde{\boldsymbol{\rho}}'^{\mathrm{u}} \\ \delta\mathbf{b}'^{\mathrm{u}} \\ \delta\mathbf{w}^{\mathrm{u}} \end{pmatrix}.
\end{aligned} \tag{27}
$$

Using Eq. (22) and Eq. (27) it may be confirmed that $\mathbf{U}_{\mathrm{p}}\mathbf{U}_{\mathrm{p}}^{-1} = \mathbf{I}$. The adjoint of Eq. (27) is constructed directly from the code.

### 4.2.3 The inverse spatial transform, $\mathbf{U}_s^{-1}$

In the current configuration, the spatial transform comprises separate horizontal ($\mathbf{U}_h$), vertical ($\mathbf{U}_v$), and scaling ($\mathbf{\Sigma}$) trans-
forms. The order of these transforms may vary. The first ordering is called the 'classic transform order' (CTO, since this was
the transform order in the first Met Office Var system, Wlasak and Cullen (2014)):

$$\mathbf{U}_s^{-1} = \mathbf{U}_h^{-1}\mathbf{U}_v^{-1}\mathbf{\Sigma}^{-1}, \tag{28}$$

and the second is called the 'reversed transform order' (RTO):

$$\mathbf{U}_s^{-1} = \mathbf{U}_v^{-1}\mathbf{U}_h^{-1}\mathbf{\Sigma}^{-1}. \tag{29}$$

$\mathbf{\Sigma}$ is a diagonal matrix of background error standard deviations of the parameters, as a function of longitude and height
(although options are implemented to allow the standard deviation to be a function of height only, or a constant for each
parameter). After the parameters have been divided by $\mathbf{\Sigma}$, the problem remains one of modelling the covariances between
spatial points in space.

There are separate spatial operators for each parameter defined in Sect. 4.2.1, and so strictly we should define the overall
spatial transforms as block-diagonal forms, but we instead adopt a casual way of describing the transforms to avoid getting
bogged down in notation. Depending on the context, these transforms may represent all parameters at once (as done in Sect.
3.4), single parameters, or to individual horizontal levels or vertical columns (as is done below).

$\mathbf{U}_h$ and $\mathbf{U}_v$ have the same generic form, as follows:

$$\mathbf{U}_{h/v}^{-1} = \mathbf{G}\mathbf{\Lambda}^{-1/2}\mathbf{F}^T, \tag{30a}$$

$$\text{so } \mathbf{U}_{h/v} = \mathbf{F}\mathbf{\Lambda}^{1/2}\mathbf{G}^T, \tag{30b}$$

where $\mathbf{F}$ is the (exact or assumed) matrix of eigenvectors (columns of $\mathbf{F}$) of the covariance matrix that is being modelled, $\mathbf{\Lambda}^{1/2}$
is the diagonal matrix of eigenvalues, and $\mathbf{G}$ is any orthonormal square matrix ($\mathbf{G}^T\mathbf{G} = \mathbf{I}$) of the same dimensions as $\mathbf{\Lambda}^{1/2}$.
In Eq. (30a), $\mathbf{F}^T$ projects a state onto the eigenvectors (mutually uncorrelated by definition), $\mathbf{\Lambda}^{-1/2}$ scales the projections so
they have have unit variance, and $\mathbf{G}$ is an arbitrary rotation. If the complete CVT had this form (but also incorporating $\mathbf{\Sigma}$),
then the implied covariance would, by Eq. (10), be $\mathbf{\Sigma}\mathbf{F}\mathbf{\Lambda}^{1/2}\mathbf{G}^T\left(\mathbf{\Sigma}\mathbf{F}\mathbf{\Lambda}^{1/2}\mathbf{G}^T\right)^T = \mathbf{\Sigma}\mathbf{F}\mathbf{\Lambda}^{1/2}\mathbf{G}^T\mathbf{G}\mathbf{\Lambda}^{1/2}\mathbf{F}^T\mathbf{\Sigma} = \mathbf{\Sigma}\mathbf{F}\mathbf{\Lambda}\mathbf{F}^T\mathbf{\Sigma}$,
where $\mathbf{F}\mathbf{\Lambda}\mathbf{F}^T$ is the eigenvalue decomposition of the covariance matrix in question. In this illustration, the CVT is an exact
representation of the covariances, but when this procedure is applied in practice it is only an approximate covariance model, e.g.
due to the separation of the horizontal and vertical transforms or to the application of approximate eigenvectors. The structure
of the actual implied covariances can be investigated with the software suite (Sect. 4.4).

In the ABC-DA system, we use the following for $\mathbf{F}$ and $\mathbf{G}$ in Eq. (30).

- For the horizontal transform, $\mathbf{U}_h^{-1}$, $\mathbf{F}$ is called $\mathbf{F}_h$ whose columns comprise horizontal plane waves of the form $\sim$
  $\exp ikx$, where $i = \sqrt{-1}$ and $k$ is the wavenumber (each column of $\mathbf{F}_h$ is a different $k$). In this context, $\mathbf{F}_h^T$ represents
  a horizontal Fourier transform. This makes the assumption that the eigenvectors of the horizontal covariance matrix are

plane waves, and the eigenvalues in $\Lambda_h$ are their variances. This is equivalent to assuming horizontal error covariances

that are homogeneous (see e.g. Bartello and Mitchell (1992); Berre (2000); Bannister (2008b)). $\mathbf{G}$ is set to $\mathbf{I}$ in the

horizontal transform.

– For the vertical transform, $\mathbf{U}_v^{-1}$, $\mathbf{F}$ is called $\mathbf{F}_v$ whose columns are the eigenmodes of a vertical covariance matrix (labelled with $\nu$, see below), and $\Lambda_v$ are their variances. $\mathbf{G}$ is set either to $\mathbf{F}_v$ to give a symmetric vertical transform, or to $\mathbf{I}$, depending on user choice.

The spaces that these operators work in depends on the chosen order of the transforms and on whether the vertical transform is symmetric or not. The following summarises these options and is repeated for each parameter, and can be read with Fig. 1 which shows how the transforms change the horizontal and vertical co-ordinates.

– For the CTO, $\mathbf{U}_v^{-1}$ operates vertically on a field that is a function of $x$ and $z$. The vertical eigenvectors/values are those of a pre-computed horizontally averaged vertical covariance matrix, and so these matrices themselves are not dependent

on horizontal position in ABC-DA.

– For the symmetric vertical transform option, $\mathbf{U}_v^{-1} = \mathbf{F}_v \Lambda_v^{-1/2} \mathbf{F}_v^T$, the output of $\mathbf{U}_v^{-1}$ is also a field that is a function of $x$ and $z$. The horizontal transform, $\mathbf{U}_h^{-1}$, then operates horizontally on such a field. The horizontal eigenvalues are those of pre-computed horizontal covariance matrices (one for each $z$ in ABC-DA). The output of $\mathbf{U}_h^{-1}$ is a field that is a function of $k$ and $z$ (see Fig. 1a). This combination of options allows a different horizontal covariance

to be specified for each vertical level.

– For the non-symmetric vertical transform option, $\mathbf{U}_v^{-1} = \Lambda_v^{-1/2} \mathbf{F}_v^T$, the output of $\mathbf{U}_v^{-1}$ is a field that is a function of $x$ and vertical eigenmode index, $\nu$. The horizontal transform, $\mathbf{U}_h^{-1}$, then operates horizontally on such a field. The horizontal eigenvalues are those of pre-computed horizontal covariance matrices (one for each $\nu$ in ABC-DA). The output of $\mathbf{U}_h^{-1}$ is a field that is a function of $k$ and $\nu$ (see Fig. 1b). This combination of options allows a

different horizontal covariance to be specified for each vertical mode, effectively allowing horizontal and vertical length-scales to be associated.

– For the RTO, $\mathbf{U}_h^{-1}$, operates horizontally on a field that is a function of $x$ and $z$. The horizontal eigenvalues are those of pre-computed horizontal covariance matrices (one for each $z$ in ABC-DA). The output of $\mathbf{U}_h^{-1}$ is a field that is a function of $k$ and $z$. The vertical transform, $\mathbf{U}_v^{-1}$, uses a separate set of vertical eigenvectors/values for each $k$.

– For the symmetric vertical transform option, the output of $\mathbf{U}_v^{-1}$ is also a field that is a function of $k$ and $z$ (see Fig. 1c).

– For the non-symmetric vertical transform option, the output of $\mathbf{U}_v^{-1}$ is a field that is a function of $k$ and $\nu$ (see Fig. 1d).

We would expect no difference between the implied covariances of the symmetric and non-symmetric vertical transform options

in the reversed case, although both options exist in the code.

**Figure 1.** Schema to illustrate the different options for the spatial transforms as indicated by the panel titles (these are combinations of classic and reversed transform orders, and symmetric and non-symmetric vertical transforms). Representations of the vertical direction include model levels (labeled with $z_1$, $z_2$, etc.) and vertical modes ($\nu_1$, $\nu_2$, etc.). Representations of the horizontal direction include model grid-points ($x_1$, $x_2$, etc.) and Fourier modes ($k_1$, $k_2$, etc.). Moving from left-to-right indicates the forward transform, and from right-to-left indicates the inverse transform. The horizontal transform is always done independently for each vertical co-ordinate ($z_i$-to-$z_i$ or $\nu_i$-to-$\nu_i$), and the vertical transform is always done independently for each horizontal co-ordinate ($x_i$-to-$x_i$ or $k_i$-to-$k_i$, as guided by the colours). The co-ordinates used in the control space in each option are indicated on the left-most panels.

### 4.2.4 The forward spatial transform, $\mathbf{U_s}$

The forward spatial transforms follow in a straightforward way from the inverses defined in Sect. 4.2.3, namely for the CTO:

$$\mathbf{U_s} = \mathbf{\Sigma}\mathbf{U_v}\mathbf{U_h}, \tag{31}$$

and for the RTO:

$$\mathbf{U_s} = \mathbf{\Sigma}\mathbf{U_h}\mathbf{U_v}. \tag{32}$$

The adjoint operators follow in a straightforward manner.

## 4.3 Calibrating the CVTs (B-matrix)

The CVTs comprise many sub-matrices that need to be determined in a calibration procedure. The operators to be determined are the regression operator $\mathbf{R}_\rho$ (part of the parameter transform mentioned in Sects. 4.2.1 and 4.2.2), and $\mathbf{\Sigma}$, $\mathbf{\Lambda_h}$, $\mathbf{F_v}$, and $\mathbf{\Lambda_v}$
(parts of the spatial transforms mentioned in Sects. 4.2.3 and 4.2.4). The number of pieces of information to be determined in this procedure is explored in Appendix A. These matrices are determined from model training data in five stages, all using the program *Master_Calibration* (*GitHubDoc§4.4*) and they are stored in a covariance file, which is produced by this routine. It is impossible to use a genuine sample of forecast errors to calibrate the $\mathbf{B}$-matrix, so instead we use ensembles of forecast perturbations, which are considered proxies of forecast error (Buehner, 2005; Pereira and Berre, 2006). The five calibration
stages are described here, and example outputs are shown for a standard set-up (experiment GB+VR+ to be described in Sect. 5.1).

### 4.3.1 Generate a population of training data from UM fields

This is calibration run stage 1 (*Master_Calibration* is run with the namelist variable *CalibRunStage* set to 1). This takes data from one or more UM files (one or more ensembles of forecasts) and extracts multiple longitude/height slices from these files
to construct an effective 'super ensemble'. These are each adjusted to make them compatible with the ABC model (as in Sect. 4.1) followed by a short forecast of specified length (in our examples one hour). These procedures are intended to give the ensemble members properties of the ABC model rather than the Unified Model from which they came, although the degree to which this has been achieved is not demonstrated. The super ensemble is output from this stage. Also specified at this stage are the model parameters (in the example to be described, $A = 0.02$ s$^{-1}$, $B = 0.01$, and $C = 10000$ m$^2$s$^{-2}$), and user settings for
the transform options mentioned in Sect. 4.2 (here, unless stated otherwise, the control options use GB ($\alpha = 1$), Sect. 4.2.1 step 3; VR, step 4; HB ($\beta = 1$), step 6; no anelastic balance ($\gamma = 0$), step 8; the CTO, Sect. 4.2.3; non-symmetric vertical transform, Sect. 4.2.3; and parameter standard deviations that are a function of vertical level only). These are all output in a provisional covariance file (netCDF format). At this stage this file is blank apart from containing information on these options for future reference. These user options are read from this file in later stages of the calibration where the above mentioned matrices are
computed and output to this covariance file. Technical information is given in *GitHubDoc§4.4.1* and the ensemble members can be plotted using the Python program specified there.

In this paper, a super ensemble of 260 members is used. Appendix A shows that this is more than adequate to determine the spatial transforms matrices and the vertical regression matrix.

### 4.3.2 Generate a population of forecast perturbations

This is calibration run stage 2 (*Master_Calibration* is run with the namelist variable *CalibRunStage* set to 2). The forecasts output from stage 1 are converted to means and perturbations from the means. See *GitHubDoc§4.4.2*, which includes plotting information.

### 4.3.3 Compute the vertical regression matrix $\mathbf{R}_\rho$

This is calibration run stage 3 (*Master_Calibration* is run with the namelist variable *CalibRunStage* set to 3). The perturbations from stage 2 are used to calculate populations of $\delta\tilde{\rho}'^{\mathrm{b}}$ Eq. (15). The vertical correlations between $\delta\tilde{\rho}'^{\mathrm{b}}$ with itself, and $\delta\tilde{\rho}'$ with $\delta\tilde{\rho}'^{\mathrm{b}}$ are then used to compute $\mathbf{R}_\rho$ in the way specified in point 4 of 4.2.1 (see also Appendix B). $\mathbf{R}_\rho$ is then output to the covariance file created in stage 1. See *GitHubDoc§4.4.3*, which includes plotting information.

Panels (a) and (b) of Fig. 2 compare example $\delta\tilde{\rho}'$ and $\delta\tilde{\rho}'^{\mathrm{b}}$ fields. The large-scale pattern of these fields is similar, with $\delta\tilde{\rho}'^{\mathrm{b}}$ being smoother and lower magnitude than $\delta\tilde{\rho}'$, indicating that the unbalanced contributions to $\delta\tilde{\rho}'$ are at smaller scales (as expected). Panel (c) shows an example $\mathbf{R}_\rho$ matrix and panel (d) shows its effect on $\delta\tilde{\rho}'^{\mathrm{b}}$, showing its ability to modify values and vertical scales.

### 4.3.4 Perform the inverse parameter transform on the forecast perturbations

This is calibration run stage 4 (*Master_Calibration* is run with the namelist variable *CalibRunStage* set to 4). The perturbations from stage 2 are transformed to parameters using the procedure represented by $\mathbf{U}_{\mathrm{p}}^{-1}$ (Sect. 4.2.1), and then output. The mean states found from stage 2 are also used in some of the calculations, e.g. $\tilde{\rho}_0$ is used in step 8 of that procedure. See *GitHubDoc§4.4.4*, which includes plotting information.

### 4.3.5 Calibrate the spatial transforms for each parameter

This is calibration run stage 5 (*Master_Calibration* is run with the namelist variable *CalibRunStage* set to 5). The perturbations from stage 4 are used to diagnose the matrices $\mathbf{\Sigma}$, $\mathbf{F}_{\mathrm{v}}$, $\mathbf{\Lambda}_{\mathrm{v}}$, and $\mathbf{\Lambda}_{\mathrm{h}}$ for each of the five control parameters.

The blue lines of Fig. 3 show the background error standard deviations (diagonal elements of $\mathbf{\Sigma}$) for the control experiment described above (see the Fig. caption for a succinct summary). The parameters show some variability with height, although none of the parameters shows variations of $\mathbf{\Sigma}$ of orders of magnitude. Note that the heights in the ABC model do not correspond with those in the real atmosphere[2]. The large variability of $w$ (panel e) at model heights around 13/14 km for instance does not

---

[2]This is the case because, for simplicity, the irregularly-spaced UM model levels are assigned new regularly-spaced heights in the ABC model when generating training data in Sect. 4.3.1.

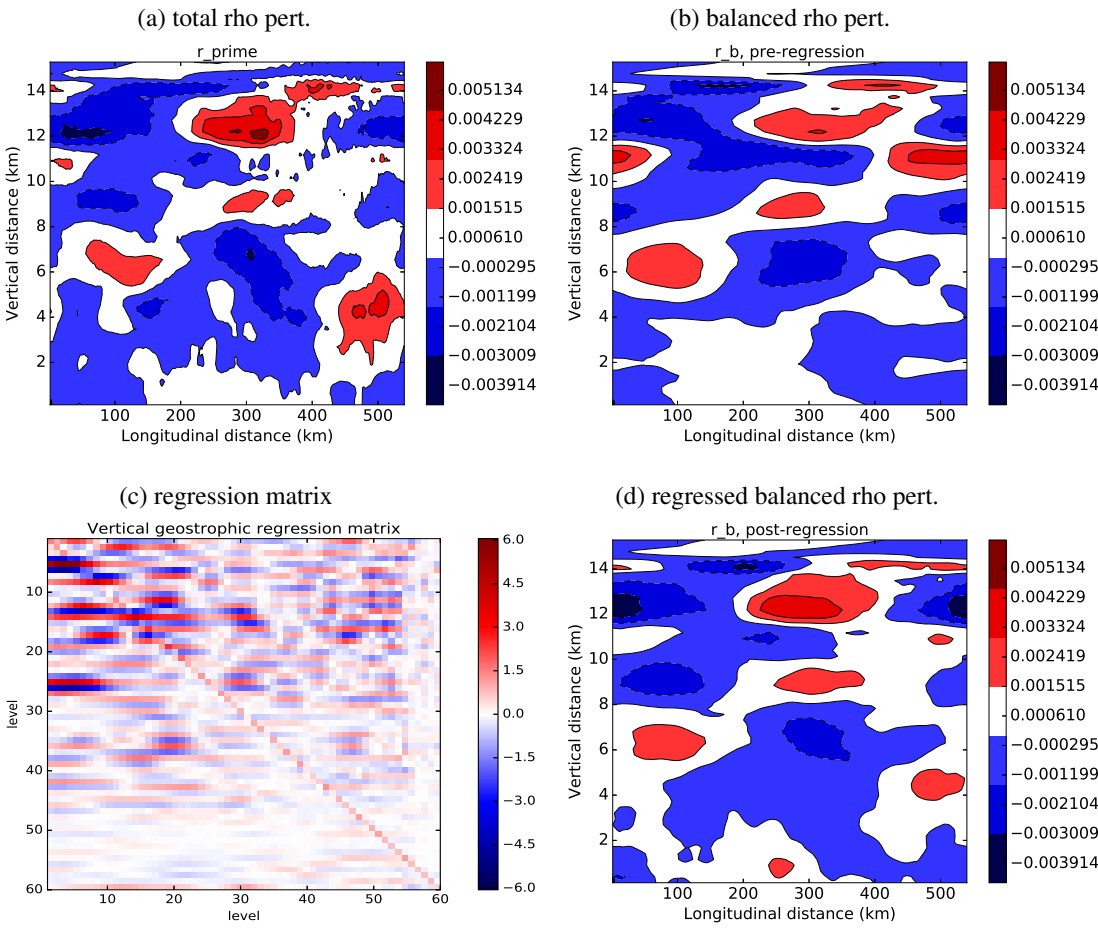

**Figure 2.** Plots comparing an example total scaled density perturbation, $\delta\tilde{\boldsymbol{\rho}}'$, with the diagnosed balanced part, $\delta\tilde{\boldsymbol{\rho}}'^{\mathrm{b}}$; and showing the effect of the regression matrix. Panel (a): $\delta\tilde{\boldsymbol{\rho}}'$ (output from stage 2 of the calibration procedure), panel (b): $\delta\tilde{\boldsymbol{\rho}}'^{\mathrm{b}}$ (diagnosed from the stream-function as in Eq. (15)), panel (c): the regression matrix $\mathbf{R}_\rho$ (found from stage 3), and panel (d): its effect on $\delta\tilde{\boldsymbol{\rho}}'^{\mathrm{b}}$, i.e. $\delta\tilde{\boldsymbol{\rho}}'^{\mathrm{br}} = \mathbf{R}_\rho\delta\tilde{\boldsymbol{\rho}}'^{\mathrm{b}}$. Note that in (c) the lowermost level corresponds to the top of the matrix.

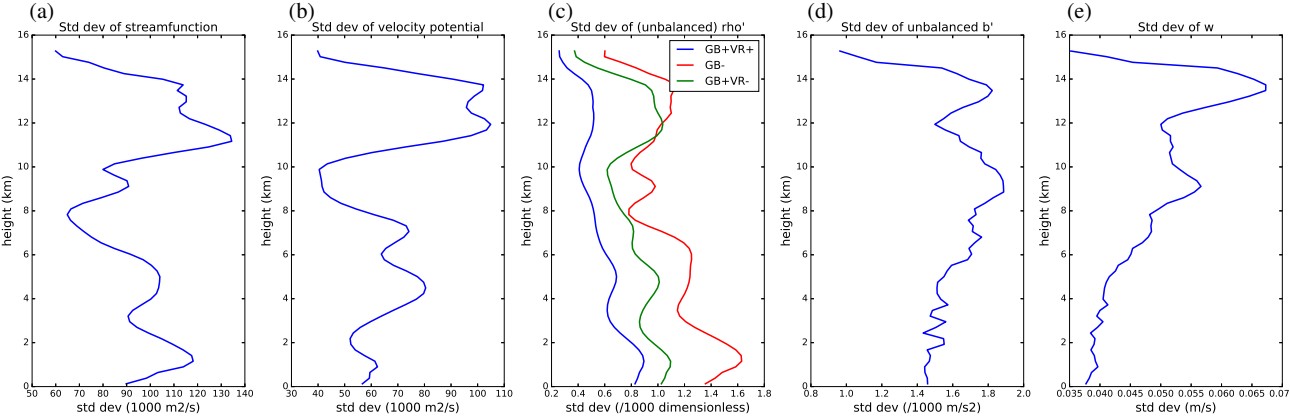

**Figure 3.** Profiles of background error standard deviations for the five control parameters with height: (a) stream-function $\delta\boldsymbol{\psi}$, (b) velocity potential $\delta\boldsymbol{\chi}_{\mathrm{vp}}$, (c) unbalanced scaled density $\delta\tilde{\boldsymbol{\rho}}'^{\mathrm{u}}$, (d) unbalanced buoyancy $\delta\mathbf{b}'^{\mathrm{u}}$, and (e) vertical wind $\delta\mathbf{w}$. The blue lines are for the experiment described in the text – namely GB and VR are switched on in the parameter transform. In panel (c) the red line is for the experiment with GB (and hence VR) switched off, and the green line is for the experiment with GB switched on and VR switched off. In the other panels all experiments yield the same profiles. The values have been smoothed using a running average over the nearest five levels.

lie in the ABC model's stratosphere, and in any case without radiative forcing in the ABC model and an ozone layer, we would
not expect signatures of the stratosphere to be present.

The blue lines of Fig. 4 show the square-root of the eigenvalues of the vertical covariance matrix (diagonal elements of $\boldsymbol{\Lambda}_{\mathrm{v}}^{1/2}$)[3]. We find the vertical covariance matrices of each parameter over each super ensemble member, and over each longitude. Each $y$-axis in Fig. 4 is the (integer) vertical model index. The eigensolver sorts these into ascending value of eigenvalue and so the physical meaning of the modes can be unclear. For instance examining the eigenvectors by eye (not shown), for parameters
$\delta\boldsymbol{\psi}$ (a), $\delta\boldsymbol{\chi}_{\mathrm{vp}}$ (b), and $\delta\tilde{\boldsymbol{\rho}}'^{\mathrm{u}}$ (c), the vertical modes of low index have vertical profiles that are generally rapidly oscillating, and have more weight in the lower model levels than in the upper ones. As the vertical mode index increases, the oscillations become less rapid and tend to have weight over the entire depth of the model atmosphere. There is no obvious trend concerning the vertical modes of $\delta\mathbf{b}'^{\mathrm{u}}$ (d) and $\delta\mathbf{w}$ (e). The values in Fig. 4 are of comparable magnitude between parameters because the vertical error covariance matrices are formed from the populations after they have been normalized with $\boldsymbol{\Sigma}^{-1}$ – see Eq. (31)
and Eq. (32).

Calibrating the vertical transform involves (for the CTO) constructing a single global vertical covariance matrix or (for the RTO) constructing one vertical covariance matrix for each wavenumber. The eigenvalues and eigenvectors follow from this procedure. The eigenvectors of the horizontal covariance matrix though are assumed to be plane waves. In the case of the CTO with a non-symmetric vertical covariance model for example, the horizontal variances follow from passing each ensemble

---

[3]In principle the vertical covariance matrix should actually be a correlation (rather than covariance) matrix given that the populations have been divided by $\boldsymbol{\Sigma}$. Due to the approximations made, the diagonal elements of this matrix may not be exactly unity.

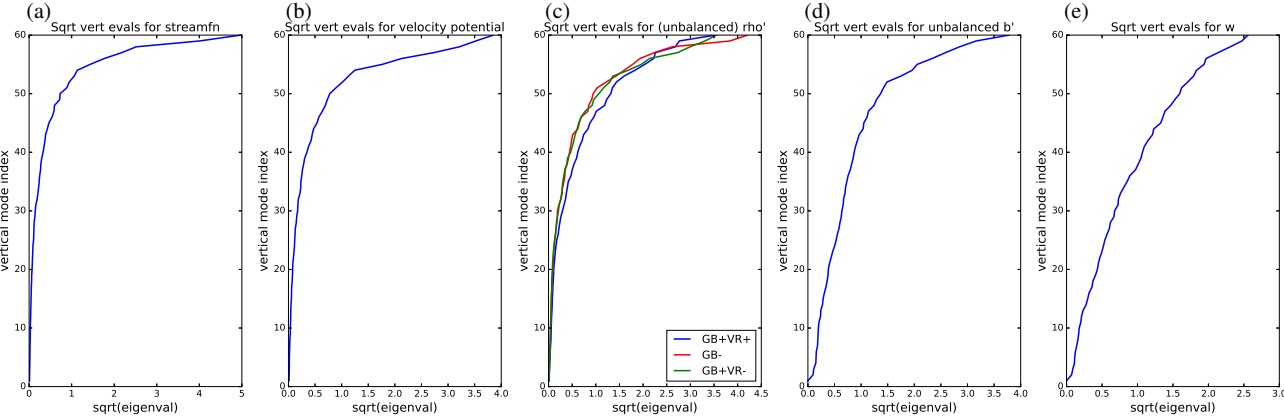

**Figure 4.** Profiles of the square-root of the vertical eigenvalues for the five control parameters as a function of vertical mode index, $\nu$: (a) $\delta\boldsymbol{\psi}$, (b) $\delta\boldsymbol{\chi}_{\mathrm{vp}}$, (c) $\delta\tilde{\boldsymbol{\rho}}'^{\mathrm{u}}$, (d) $\delta\mathbf{b}'^{\mathrm{u}}$, and (e) $\delta\mathbf{w}$. The blue lines are for the experiment described in the text – namely GB and VR switched on. In panel (c) the red line is for the experiment with GB (and hence VR) switched off, and the green line is for the experiment with GB switched on and VR switched off. In the other panels all experiments yield the same profiles.

perturbation ($\delta\mathbf{x}$) through the partial inverse transform to give $\delta\boldsymbol{\eta} = \mathbf{F}_{\mathrm{h}}^{\mathrm{T}}\mathbf{U}_{\mathrm{v}}^{-1}\boldsymbol{\Sigma}^{-1}\mathbf{U}_{\mathrm{p}}^{-1}\delta\mathbf{x}$. Here each $\delta\boldsymbol{\eta}$ is a field which is function of $k$ and $\nu$. This ensemble is then used to estimate the variances of $\delta\boldsymbol{\eta}$, $\boldsymbol{\Lambda}_{\mathrm{h}}^{0}(k,\nu)$.

Figure 5 shows the square-root of the horizontal variances (diagonal elements of $\boldsymbol{\Lambda}_{\mathrm{h}}^{1/2}$). In the particular configuration that has a non-symmetric vertical transform, there is a different horizontal variance spectrum for each vertical mode, so the axes in Fig. 5 are wavenumber, $k$, and vertical model index, $\nu$. The spectra for $\delta\boldsymbol{\psi}$ (a) and $\delta\boldsymbol{\chi}_{\mathrm{vp}}$ (b) are similar and represent
perturbations with long horizontal correlation length-scales (most weight is at small wavenumbers). Their similarities are explained by the fact that the fields are related to the horizontal winds in the same way: $v = \partial\delta\psi/\partial x$ and $u = \partial\delta\chi_{\mathrm{vp}}/\partial x$. The spectra for $\delta\tilde{\rho}'^{\mathrm{u}}$ (c) show that perturbations in this parameter have shorter horizontal length-scales than in (a) and (b), and the length-scale reduces with higher vertical modes. This suggests that perturbations with broader vertical scales (generally higher vertical mode index) tend to have shorter horizontal scales. The spectra for the remaining parameters, $\delta\mathbf{b}'^{\mathrm{u}}$ (d) and $\delta\mathbf{w}$ (e) show
that perturbations of these parameters are courser still and there is no obvious pattern with vertical mode index.

The same data plotted in Fig. 5c (for $\delta\tilde{\boldsymbol{\rho}}'^{\mathrm{u}}$) are also plotted in Fig. 6a, but with transformed co-ordinates to facilitate further interpretation. In Fig. 6a the variance square-roots are shown as a function of horizontal length-scale, $l_{\mathrm{h}}$, and vertical length-scale, $l_{\mathrm{v}}$. $l_{\mathrm{h}}$ is related to horizontal wavenumber index, $k$ (an integer), as $l_{\mathrm{h}} = L_{\mathrm{h}}/k$ (where $L_{\mathrm{h}} = 540$ km is the horizontal domain length, and excluding $k = 0$ since it gives an infinite $l_{\mathrm{h}}$), and $l_{\mathrm{v}}$ is related to the vertical eigenmode index for $\delta\tilde{\rho}'^{\mathrm{u}}$ in
the way outlined in Appendix C. For reference, panel b shows $l_{\mathrm{v}}$ for each vertical mode of $\delta\tilde{\boldsymbol{\rho}}'^{\mathrm{u}}$, which confirms that higher $\nu$ is broadly associated with deeper structures. Furthermore, modes with deeper structures have fewer nodes (not shown). Incidentally this correspondence between $\nu$ and $l_{\mathrm{v}}$ is also found for the vertical modes of $\delta\psi$, which represent the balanced scaled density increments.

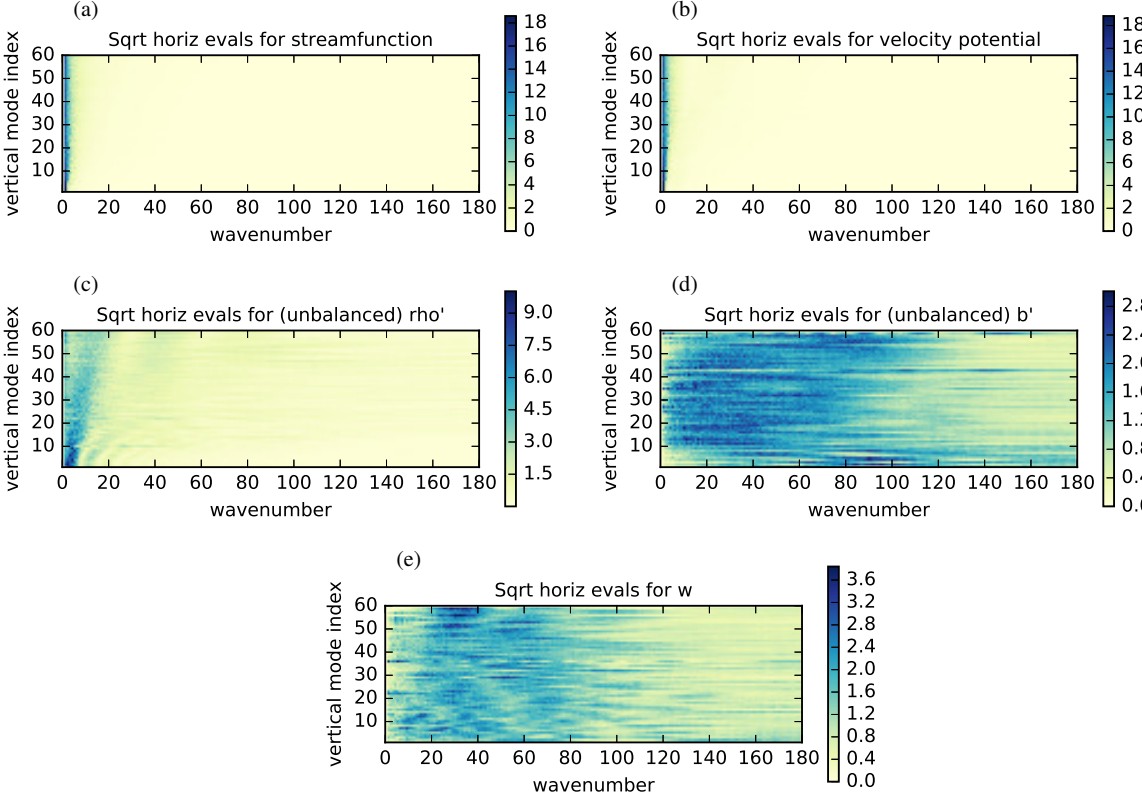

**Figure 5.** The square-root of the horizontal variances (colours) as a function of horizontal wavenumber and vertical mode index for the five control parameters: (a) $\delta\boldsymbol{\psi}$, (b) $\delta\boldsymbol{\chi}_{\mathrm{vp}}$, (c) $\delta\tilde{\boldsymbol{\rho}}'^{\mathrm{u}}$, (d) $\delta\mathbf{b}'^{\mathrm{u}}$, and (e) $\delta\mathbf{w}$.

Returning to panel a, the Rossby radius of deformation, $L_{\mathrm{R}}$, has also been plotted (the black curve). This has been estimated using a shallow water interpretation, i.e. $L_{\mathrm{R}} = \sqrt{g l_{\mathrm{v}}}/f$ (e.g. Cullen (2006)). The Burger number, $Bu$, is defined as $Bu = L_{\mathrm{R}}/l_{\mathrm{h}}$, and it follows that all of the represented variability in these unbalanced modes has large $Bu$. Dynamical theory says that for $l_{\mathrm{h}} \ll L_{\mathrm{R}}$ (i.e. $Bu \gg 1$) the rotational wind is a potential vorticity (PV)-like variable and for $l_{\mathrm{h}} \gg L_{\mathrm{R}}$ ($Bu \ll 1$) the mass variable becomes a PV-like variable (e.g. Cullen (2003); Katz et al. (2011)). PV is often associated with balanced flow, and so the above theory suggests that the unbalanced mass variable ($\delta\tilde{\rho}'^{\mathrm{u}}$) should vanish – and hence have vanishing variance – for $Bu \ll 1$. Figure 6a shows that for $10^1$ km $\lesssim l_{\mathrm{h}} \lesssim 10^3$ km, the variance of $\delta\tilde{\rho}'^{\mathrm{u}}$ does increase as $Bu$ increases, showing some consistency with the dynamical theory at synoptic scales. It further suggests that a rotational wind variable, like $\delta\psi$ is a reasonable variable to capture the balanced part of the flow at synoptic scales.

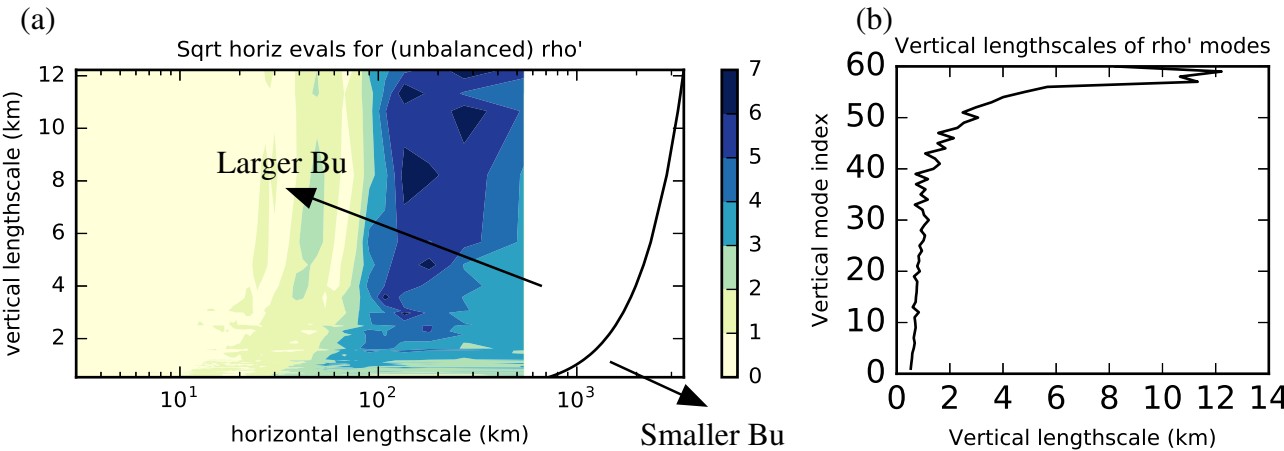

**Figure 6.** (a) The same horizontal spectra of Fig. 5c but shown as a function of horizontal ($l_\mathrm{h}$) and vertical ($l_\mathrm{v}$) length-scales. Also shown in (a) is the Rossby radius of deformation, $L_\mathrm{R}$, plotted as a function of $l_\mathrm{v}$ (black line). The Burger number is defined as $Bu = L_\mathrm{R}/l_\mathrm{h}$. (b) Vertical length-scales ($x$-axis) of the vertical modes ($y$-axis) for the unbalanced scaled density control parameter, calculated according to Appendix C.

### 4.4 The implied B-matrix

The covariance model is completely described by (i) the parameter transform presented in Sect. 4.2.2, (ii) the results presented in Sect. 4.3.5, and (iii) other statistics like the vertical eigenvectors themselves (not shown). The software suite has a program to compute the implied covariances (*Master_ImpliedCov*, see *GitHubDoc§4.6*) which uses information in the CVT file output by the calibration procedures. The program acts with the combination of matrices $\mathbf{U}\mathbf{U}^\mathrm{T}$ on a vector which is zero apart from one element (which we call the source point), which is set to unity (in the example below, this element corresponds to $\delta\tilde{\rho}'$ near the centre of the domain). Completing the matrix algebra, the implied super matrix (for fields $\delta\mathbf{u}$, $\delta\mathbf{v}$, $\delta\mathbf{w}$, $\delta\tilde{\rho}'$, $\delta\mathbf{b}'$) is found from Eq. (10), $\mathbf{U} = \mathbf{U}_\mathrm{p}\mathbf{U}_\mathrm{s}$, and Eq. (27):

$$
\begin{pmatrix}
\mathbf{B}_u^{\mathrm{ic}} & 0 & \gamma\mathbf{B}_u^{\mathrm{ic}}\mathbf{L}_u^{\mathrm{ab}\,\mathrm{T}} & 0 & 0 \\
\nabla_x\mathbf{B}_\mathrm{s}^\psi\nabla_x^\mathrm{T} & \alpha\gamma\frac{f}{C}\nabla_x\mathbf{B}_\mathrm{s}^\psi\mathbf{R}_\rho^\mathrm{T}\mathbf{L}_{\tilde{\rho}'}^{\mathrm{ab}\,\mathrm{T}} & \alpha\frac{f}{C}\nabla_x\mathbf{B}_\mathrm{s}^\psi\mathbf{R}_\rho^\mathrm{T} & \alpha\beta\frac{f}{C}\nabla_x\mathbf{B}_\mathrm{s}^\psi\mathbf{R}_\rho^\mathrm{T}\mathbf{L}^{\mathrm{hb}\,\mathrm{T}} \\
 & \mathbf{B}_w^{\mathrm{ic}} & \alpha^2\gamma\frac{f^2}{C^2}\mathbf{L}_{\tilde{\rho}'}^{\mathrm{ab}}\mathbf{B}_\mathrm{s}^{\psi_\mathrm{r}} + \gamma\mathbf{L}_{\tilde{\rho}'}^{\mathrm{ab}}\mathbf{B}_\mathrm{s}^{\tilde{\rho}'\mathrm{u}} & \alpha^2\beta\gamma\frac{f}{C}\mathbf{L}_{\tilde{\rho}'}^{\mathrm{ab}}\mathbf{B}_\mathrm{s}^{\psi_\mathrm{r}}\mathbf{L}^{\mathrm{hb}\,\mathrm{T}} + \gamma\beta\mathbf{L}_{\tilde{\rho}'}^{\mathrm{ab}}\mathbf{B}_\mathrm{s}^{\tilde{\rho}'\mathrm{u}}\mathbf{L}^{\mathrm{hb}\,\mathrm{T}} \\
 & & \alpha^2\frac{f^2}{C^2}\mathbf{B}_\mathrm{s}^{\psi_\mathrm{r}} + \mathbf{B}_\mathrm{s}^{\tilde{\rho}'\mathrm{u}} & \alpha^2\beta\frac{f^2}{C^2}\mathbf{B}_\mathrm{s}^{\psi_\mathrm{r}}\mathbf{L}^{\mathrm{hb}\,\mathrm{T}} + \beta\mathbf{B}_\mathrm{s}^{\tilde{\rho}'\mathrm{u}}\mathbf{L}^{\mathrm{hb}\,\mathrm{T}} \\
 & & & \mathbf{B}_{b'}^{\mathrm{ic}}
\end{pmatrix},
$$
$$(33)$$

where to save space only the upper trianglar part of the covariance matrix is shown (the rest of the matrix is the adjoint of the corresponding transpose element). The following definitions have been made for convenience:

$$\mathbf{B}_s^p = \mathbf{U}_s^p \mathbf{U}_s^{p\,\mathrm{T}} \qquad p = \delta\boldsymbol{\psi}, \delta\boldsymbol{\chi}_{\mathrm{vp}}, \delta\tilde{\boldsymbol{\rho}}'^{\mathrm{u}}, \delta\mathbf{b}'^{\mathrm{u}}, \text{or } \delta\mathbf{w}^{\mathrm{u}}, \tag{34a}$$

$$\mathbf{B}_u^{\mathrm{ic}} = \nabla_x \mathbf{B}_s^{\chi_{\mathrm{vp}}} \nabla_x^{\mathrm{T}}, \tag{34b}$$

$$\mathbf{B}_w^{\mathrm{ic}} = \alpha^2 \gamma^2 \frac{f^2}{C^2} \mathbf{L}_{\tilde{\rho}'}^{\mathrm{ab}} \mathbf{B}_s^{\psi_{\mathrm{r}}} \mathbf{L}_{\tilde{\rho}'}^{\mathrm{ab}\,\mathrm{T}} + \gamma^2 \mathbf{L}_u^{\mathrm{ab}} \mathbf{B}_u^{\mathrm{ic}} \mathbf{L}_u^{\mathrm{ab}\,\mathrm{T}} + \gamma^2 \mathbf{L}_{\tilde{\rho}'}^{\mathrm{ab}} \mathbf{B}_s^{\tilde{\rho}'^{\mathrm{u}}} \mathbf{L}_{\tilde{\rho}'}^{\mathrm{ab}\,\mathrm{T}} + \mathbf{B}_s^{w^{\mathrm{u}}} \tag{34c}$$

$$\mathbf{B}_s^{\psi_{\mathrm{r}}} = \mathbf{R}_\rho \mathbf{B}_s^{\psi} \mathbf{R}_\rho^{\mathrm{T}}, \tag{34d}$$

$$\mathbf{B}_{b'}^{\mathrm{ic}} = \alpha^2 \beta^2 \frac{f^2}{C^2} \mathbf{L}^{\mathrm{hb}} \mathbf{B}_s^{\psi_{\mathrm{r}}} \mathbf{L}^{\mathrm{hb}\,\mathrm{T}} + \beta^2 \mathbf{L}^{\mathrm{hb}} \mathbf{B}_s^{\tilde{\rho}'^{\mathrm{u}}} \mathbf{L}^{\mathrm{hb}\,\mathrm{T}} + \mathbf{B}_s^{\mathrm{b}'^{\mathrm{u}}}. \tag{34e}$$

Recall that the factors $\alpha$, $\beta$, and $\gamma$ are set to 1 or 0 to turn on or off geostrophic, hydrostatic, or anelastic balance in the covariance model (Sect. 4.2). Equation Eq. (34a) represents the implied spatial covariances, where $p$ (parameter) is short for each control parameter listed in Eq. (34a). The diagonal blocks of Eq. (33) are the implied covariances of the model variables. The structure of the terms in Eq. (33) reveals the way that the covariances have been modelled. For example, the implied covariance of $\delta\tilde{\rho}'$ is the $4, 4$th block, $\alpha^2 \frac{f^2}{C^2} \mathbf{B}_s^{\psi_{\mathrm{r}}} + \mathbf{B}_s^{\tilde{\rho}'^{\mathrm{u}}}$, and comprises a balanced contribution (the first term, which depends upon the covariance structure of $\delta\boldsymbol{\psi}$), and an unbalanced contribution (the second term, which is independent). Other variables have a similar decomposition.

This equation is too complicated to illustrate the implied geographical covariance structure, so a selection of the implied covariances Eq. (10) are computed and shown graphically in the top row of Fig. 7. All panels represent the covariances between a source point ($\delta\tilde{\rho}'$ at the position of the cross) with (a) $\delta\tilde{\rho}'$, (b) $\delta\mathbf{v}$, and (c) $\delta\mathbf{b}'$. For comparison, the bottom row shows the respective raw sample covariances found from the population of 260 forecasts used to calibrate the covariance model (found using the program *Master_RawCov*, see *GitHubDoc§4.7*). We regard the signals contained in the raw covariances as a rough (row-rank) guide to the covariances that should ideally be modelled by the CVT.

The autocovariances in (a) have a region of positive correlations about the source point, with regions of negative correlation towards the east, west, above, and below. This broad structure is apparent in the raw covariance (j) although the implied values are two to four times larger. The covariances between $\delta\tilde{\rho}'$ and $\delta\mathbf{v}$ in (b) show dipole patterns in the horizontal due to GB imposed according to Eq. (2a). This is done in the CVT via the $2, 4$th block of Eq. (33), which spreads out the source point with $\mathbf{B}_s^{\psi} \mathbf{R}_\rho^{\mathrm{T}}$, evaluates the horizontal gradient, and then multiplies by $\alpha\frac{f}{C}$. Physically, (b) is the meridional wind response (an anti-cyclone) to a positive density perturbation at the cross, and these covariances mean that this pattern would appear in the $v$ analysis increments when assimilating a single $\tilde{\rho}'$ measurement at the cross. The pattern in (b) is evident in the raw covariance in (k), but the implied version is slightly too large. Finally, the covariances between $\delta\tilde{\rho}'$ and $\delta\mathbf{b}'$ in (c) show the hydrostatic response of the density perturbation, which is the $5, 4$th block of the CVT, $\alpha^2\beta\frac{f^2}{C^2}\mathbf{L}^{\mathrm{hb}}\mathbf{B}_s^{\psi_{\mathrm{r}}} + \beta\mathbf{L}^{\mathrm{hb}}\mathbf{B}_s^{\tilde{\rho}'^{\mathrm{u}}} = \beta\mathbf{L}^{\mathrm{hb}}\left(\alpha^2\frac{f^2}{C^2}\mathbf{B}_s^{\psi_{\mathrm{r}}} + \mathbf{B}_s^{\tilde{\rho}'^{\mathrm{u}}}\right)$. There are two contributions to the $\delta\tilde{\rho}'$-$\delta\mathbf{b}'$ covariances: one involves the route $\delta\tilde{\rho} \to \delta\psi \to \delta\mathbf{b}'$ (exploiting GB and HB), and the other involves the route $\delta\tilde{\rho} \to \delta\tilde{\rho}'^{\mathrm{u}} \to \delta\mathbf{b}'$ (exploiting only HB). A closer inspection of Eq. (33) reveals that the $5, 4$th block is equal to $\beta\mathbf{L}^{\mathrm{hb}}$ (proportional to the vertical derivative operator – see line after Eq. (18)) acting on the $4, 4$th block, which is the $\delta\tilde{\rho}'$ response. Figure 7 shows that (c) is indeed related to the vertical derivative of (a).

The structure of (c) comprises complicated bands of positive and negative covariances, and again this pattern would appear in the $b'$ analysis increments when assimilating a single $\tilde{\rho}'$ measurement at the cross. The structure in (c) is evident in the raw covariances (l) suggesting that HB is followed by the forecasts to some extent although the implied covariances are up to about three times larger.

## 4.5 Observations and observation operators

ABC-DA currently supports observation operators for direct (point) observations of $u$, $v$, $w$, $\tilde{\rho}'$, $b'$, a conserved tracer, $\sqrt{u^2 + v^2}$, and $\sqrt{u^2 + v^2 + w^2}$. These correspond to observation codes 1-8 respectively. Observations are read-in via a text file, which specifies the time of each observation, its longitudinal and height position, the observation code, the observation value itself, and its error standard deviation. The format of the input file is specified in *GitHubDoc§4.8*. The observation operators employ a bi-linear interpolation and observation errors are assumed to be uncorrelated.

The user may extend the range of observation operators available by assigning a new observation code, and adding a new branch in the observation operator and adjoint routines. Furthermore, observations are assigned a batch number. This is not used in the current code, but it may be used by the user to allow for correlated errors within each batch.

## 4.6 Generating data for twin experiments

The program *Master_MakeBgObs*, see *GitHubDoc§4.8*, is available to generate an observation network, to generate synthetic
random observations given an observation network (for later assimilation), and to generate a random background state that is consistent with the background errors as specified in a CVT file. Further details are as follows.

- An observation network is a specification of a grid of observation times, locations, error standard deviations, and which synthetic observations to make (but not the observation values themselves). For example, the user may wish to generate synthetic observations of $s = \sqrt{u^2 + v^2}$ in an $N_x^s \times N_z^s$ grid between $x_1^s$ and $x_2^s$ and $z_1^s$ and $z_2^s$ and synthetic observations
of $\tilde{\rho}'$ in an $N_x^{\tilde{\rho}'} \times N_z^{\tilde{\rho}'}$ grid between $x_1^{\tilde{\rho}'}$ and $x_2^{\tilde{\rho}'}$ and $z_1^{\tilde{\rho}'}$ and $z_2^{\tilde{\rho}'}$. Running *Master_MakeBgObs* with the option to generate this observation network (specified in an input file) will produce a text file containing entries for each individual observation (where it should be made, when it should be made, which quantity should be measured, and at what precision – see *GitHubDoc§4.8*). This file could be made by hand, but this software option exists to make the task less tedious. Once produced, this file can be edited by removing, adding, or altering individual observation points.

- Pseudo observations are generated according to the output from the above procedure, and an ABC model initial condition ('truth') file. Running *Master_MakeBgObs* in this mode will run a forecast from the initial conditions, generate 'true observations' from the forecast, and will then add observation noise to each observation. The output from this stage is a file of synthetic observations which can then be assimilated (Sect. 4.7). The true observations are also output within this data file, which can be used later for error analysis. The user may wish to generate synthetic observations for an identical
twin experiment by using the same model parameters (i.e. $A$, $B$, $C$) as will be used in the DA, or may wish to run a non-identical twin experiment with different parameters. As with the procedure for generating the observation network,

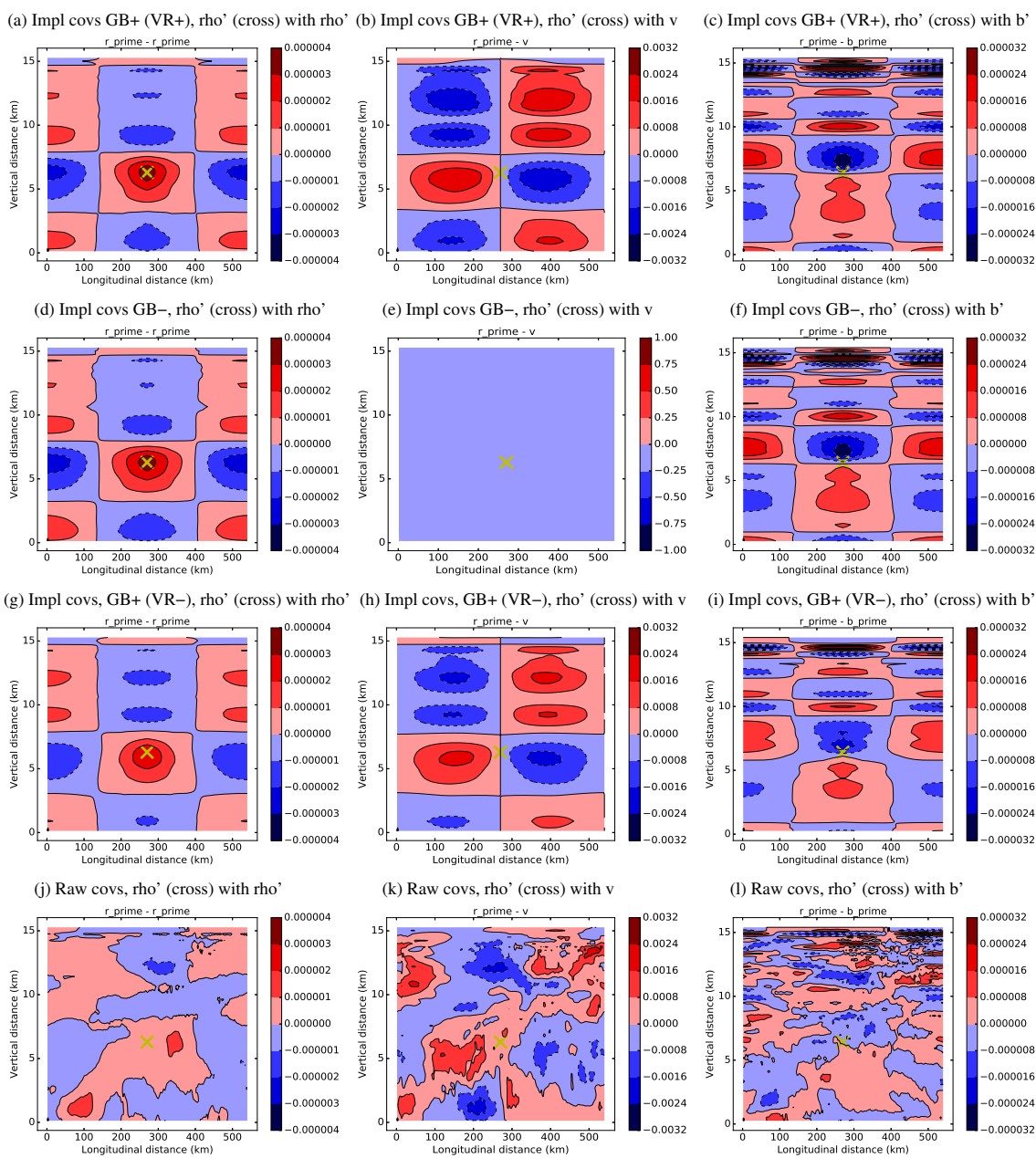

**Figure 7.** A selection of covariances (panel columns) found in different ways (rows). All covariances are between the source point ($\delta\tilde{\boldsymbol{\rho}}'$ at the cross) and $\delta\tilde{\boldsymbol{\rho}}'$ (left column), $\delta\mathbf{v}$ (middle column), and $\delta\mathbf{b}'$ (right column) at the field locations. The rows correspond to: implied covariances when GB and VR are used in the CVT (GB+VR+, top row), implied covariances when GB is not used (GB-, second row), implied covariances when GB is used and VR is not used (GB+VR-, third row), and when covariances are calculated directly from the population used to calibrate each CVT (bottom row). Red (blue) indicates positive (negative) covariances.

this file can be edited, or instead made by hand from scratch. For example, the user may want to add, remove, or change individual observations in order to investigate the effect on the assimilation.

– A valid background state may be generated with the program *Master_MakeBgObs*. The 'truth' state (conventionally that used to generate the synthetic observations) is perturbed with a random state $\delta\mathbf{x} = \mathbf{U}\delta\boldsymbol{\chi}$ Eq. (8), where $\delta\boldsymbol{\chi}$ is drawn from $N(0, \mathbf{I})$. The information used to describe the CVT, $\mathbf{U}$ is found from a specified CVT file.

## 4.7 Data assimilation

The program that performs a DA cycle is called *Master_Assimilate* , see *GitHubDoc§4.9*. The inputs to this program are: (i) a background state (e.g. as generated in Sect. 4.6), (ii) a set of observations (e.g. as generated in Sect. 4.6), and (iii) a description of the $\mathbf{B}$-matrix (in a CVT as found using the calibration procedure in Sect. 4.3). The assimilation methods currently available are 3DVar and 3DFGAT, although the code is flexible enough to allow other methods (e.g. 4DVar, EnVar, etc., Bannister (2017)) to be developed.

The algorithm for 3DFGAT (the preferred option in the current version) is shown in Fig. 8. The reference state, $\mathbf{x}^{\mathrm{r}}$, is set to the background, $\mathbf{x}^{\mathrm{b}}$, which is then used to compute the innovations, $\mathbf{d}(t)$, computed at their appropriate times within the DA window. The control space perturbation, $\delta\boldsymbol{\chi}$ is set to zero. $\mathbf{d}(t)$ and $\delta\boldsymbol{\chi}$ enter the inner loop minimisation (pink box), where the residuals, $\mathbf{r}(t)$, are computed. These are used in the calculation of $\nabla_{\delta\boldsymbol{\chi}}J_{\mathrm{o}}$, where $J_{\mathrm{o}}$ is the observation term of the cost function. This is added with the background contribution, $\nabla_{\delta\boldsymbol{\chi}}J_{\mathrm{b}}$, giving the total cost function gradient. This is used with a descent algorithm – in our case a conjugate gradient algorithm – which provides the increment, $\Delta\delta\boldsymbol{\chi}$. The switch is flipped (red arrow inside the pink box) so that the inner loop procedure is repeated with the $\delta\boldsymbol{\chi}$ that is updated with $\Delta\delta\boldsymbol{\chi}$. Once converged, the reference state is incremented by $\mathbf{U}\delta\boldsymbol{\chi}$. This ends the first outer loop iteration. If further outer loop iterations are required, the switch is flipped (red arrow outside of the pink box), so that the incremented $\mathbf{x}^{\mathrm{r}}$ is fed into the innovation calculation. The $\mathbf{x}^{\mathrm{r}}$ at the final outer loop is the analysis, $\mathbf{x}^{\mathrm{a}}$.

In addition to the analysis, and the analysis increment, there are many diagnostic outputs from the DA, as detailed in *GitHubDoc§4.9*. This includes the reference states at each outer loop, the gradient $\nabla_{\delta\mathbf{x}}J_{\mathrm{o}}$, the values of the cost function with iteration, and versions of the observations file with extra information. The last file is output for each outer loop and includes the model observation values computed from $\mathbf{x}^{\mathrm{r}}$ (the $\mathbf{x}^{\mathrm{r}}$ corresponding to the last outer loop is $\mathbf{x}^{\mathrm{a}}$), the innovation, $\mathbf{d}(t)$, the residual, $\mathbf{r}(t)$, and the gradient of the observation term in the cost function with respect to the model observation, $\mathbf{R}_t^{-1}\mathbf{r}(t)$. These diagnostics can be plotted using the python program specified in *GitHubDoc§4.9*. A selection of example diagnostics is shown in Sect. 5 in connection with the study on the effect of the GB conditions in the CVT.

## 4.8 Cycling

A (provided) bash script controls the schedule of tasks to perform DA cycling, see *GitHubDoc§5*. The background state for the first cycle is made using the procedure in 4.6, but subsequent backgrounds are found from forecasts originating from each

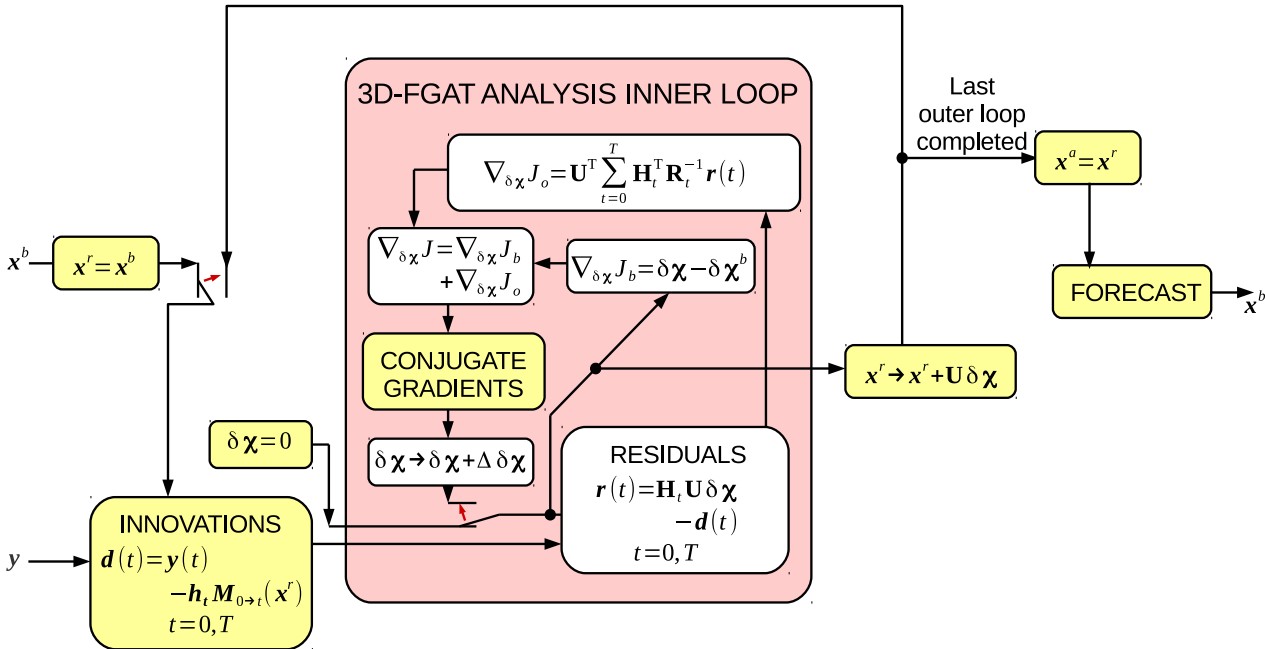

**Figure 8.** The outer and inner loops of the variational 3DFGAT procedure. The main inputs into the algorithm are the background state, $\mathbf{x}^b$, and the observations, $\mathbf{y}$. The algorithm inside the pink box is the inner loop. The small red arrows represent switches in the data flow. The initial positions (shown) are changed after the first iteration. This algorithm incorporates cycling, where the forecast made (on the right hand side) is fed back (on the left hand side) for the next assimilation window. The full 4DVar algorithm to calculate the gradient $\nabla_{\delta\chi} J$ (essentially part of the inner loop) is given in Sect. 3.5. Note that $\delta\chi^b = \mathbf{U}^{-1}\left(\mathbf{x}^b - \mathbf{x}^r\right)$.

analysis. The script also propagates the true state from one cycle to the next, for generation of the synthetic observations and for subsequent monitoring.

Summary statistics of cycling experiments, and comparative diagnostics between pairs of cycling experiments can be computed and plotted using python programs provided. The study in Sect. 5 shows many of these diagnostics, which may be regarded as examples.

# 5   An investigation of balance in modelling the B-matrix

Most Var system began their lives serving global models on relatively coarse grids. In mid- and high-latitude regions, and
when only relatively large scales are represented, a significant portion of the flow fields are related via GB and HBs (GB and HB). Most Var systems use this property to imply relationships between variables, which impact the background error covariance statistics, and a similar approach is often followed in systems that use much finer grids (as is done in Sect. 4.2 for ABC-DA). Since GB becomes a less important constraint at smaller scales (Berre, 2000; Bannister et al., 2011; Petrie et al.,

2017), the research question arises concerning whether using such a constraint in the **B**-model is necessary, or indeed harmful in convective-scale DA.

Some regional Var systems, such as those for the ALADIN (Aire Limitée Adaption Dynamique et dévelopment InterNational, Berre (2000)) and WRF (Weather Research and Forecasting model, Chen et al. (2013)) models couple mass and wind fields via a linear regression operator, trained from sample data. The degree of balance represented by these operators will be appropriate to the training sample. Other systems, such as the current configuration of ABC-DA (Sect. 4.2), the Met Office (Ingleby, 2001; Bannister, 2008b), and HIRLAM (HIgh Resolution Limited-Area Model, Gustafsson et al. (2001)) impose analytic GB operators. Analytic relationships are 'cleaner' and more physical than regressions, but their current forms are not so adaptive to the underlying conditions. Geophysical balances are not only sensitive to scale and latitude, but also to precipitation, where heavy precipitation leads to a weakening of the balance between mass and wind (e.g. Caron and Fillion (2010); Chen et al. (2016)). Few, if any, studies have examined the effect of switching off the GB constraint completely in the **B**-matrix in convective-scale systems. This is one way to affect the implied covariances ($\alpha = 1 \rightarrow 0$ in Eq. (33)). Other suggestions have been made to change the nature of the wind coupling in **B** at convective-scale. Sun et al. (2016) e.g. compared using $\psi,\chi$-based uncorrelated control parameters with $u,v$-based ones (with the associated change of the CVT). Their comparison of these two alternative sets of wind control parameters was not so straightforward as their $\psi,\chi$-based set used a linear regression–type balance with mass variables, while their $u,v$-based set did not. This did not allow a clean analysis of the effect of the change of parameters verses the effect of turning off the balance constraint.

## 5.1 Description of the DA experiments

In this section we do three DA experiments each with its own configuration as described below.

– Experiment GB+VR+ This experiment mirrors the operational configuration of the Met Office DA system. In this configuration $\delta\tilde{\boldsymbol{\rho}}'$ is decomposed into a GB part, $\delta\tilde{\boldsymbol{\rho}}'^{\mathrm{b}}$, and an unbalanced part, $\delta\tilde{\boldsymbol{\rho}}'^{\mathrm{u}}$ ($\alpha = 1$ in Eq. (27) and Eq. (33)) and uses the VR operator, $\mathbf{R}_\rho$, to modify the calculation of the balanced scaled density perturbation increments from the streamfunction increment. $\delta\mathbf{b}'$ is decomposed into an HB part, $\delta\mathbf{b}'^{\mathrm{b}}$, and an unbalanced part, $\delta\mathbf{b}'^{\mathrm{u}}$ ($\beta = 1$). $\delta\mathbf{w}$ is analysed as though it were completely unbalanced ($\gamma = 0$). The ordering of the vertical and horizontal transforms is the CTO, and the non-symmetric vertical transform is used. As a reminder, the implied covariances for GB+VR+ are shown in the top row of Fig. 7.

– Experiment GB- This experiment differs from GB+VR+ in that there is no balanced component of $\delta\tilde{\boldsymbol{\rho}}'$ ($\alpha = 0$, $\beta = 1$, and $\gamma = 0$). Since there is no GB operator, the VR step is irrelevant. The implied covariances for GB- are shown in the second row of Fig. 7. When calibrated with this configuration the $\delta\tilde{\boldsymbol{\rho}}'$-$\delta\tilde{\boldsymbol{\rho}}'$ covariances (d) are almost the same as for GB+VR+ (see the $4,4$th block of Eq. (33) to see how $\alpha$ affects the $\delta\tilde{\boldsymbol{\rho}}'$-$\delta\tilde{\boldsymbol{\rho}}'$ covariances). The $\delta\tilde{\boldsymbol{\rho}}'$-$\delta\mathbf{v}$ covariances (e) are zero as expected as GB was the mechanism used to couple these two variables in the CVT (see the $2,4$th block of Eq. (33)). As $\delta\tilde{\boldsymbol{\rho}}'^{\mathrm{b}} = 0$ in this experiment, $\delta\tilde{\boldsymbol{\rho}}'^{\mathrm{u}} = \delta\tilde{\boldsymbol{\rho}}'$, which is consistent with the variances of $\delta\tilde{\boldsymbol{\rho}}'^{\mathrm{u}}$ (red line in Fig. 3(c)

for GB-) being larger than those of the blue line (GB+VR+). The $\delta\tilde{\rho}'$-$\delta\mathbf{b}'$ covariances in Fig. 7(f) are similar to those for GB+VR+ (c), showing that the effect of GB on these statistics is small.

– Experiment GB+VR- This experiment differs from GB+VR+ in that no VR is performed ($\alpha = 1$, $\beta = 1$, $\gamma = 0$, and $\mathbf{R}_\rho = \mathbf{I}$). The implied covariances for GB+VR- are shown in the third row of Fig. 7. When calibrated with this configuration the $\delta\tilde{\rho}'$-$\delta\tilde{\rho}'$ covariances (g) are slightly smaller than for GB+VR+. The effect on $\delta\mathbf{v}$ (h) is noticeably smaller than in GB+VR+. In GB+VR+, $\mathbf{R}_\rho$ has the effect of increasing the magnitude of the balanced increment ($\delta\tilde{\rho}_r'^b$ is greater than $\delta\tilde{\rho}'^b$ on average), and thus decreasing the variance of $\delta\tilde{\rho}'^u$. This is seen in the blue line of Fig. 3(c) (the variance of $\delta\mathbf{b}'^u$ in GB+VR-) being smaller than the green line (GB+VR-), and explains why the geostrophic response in Fig. 7(b) is larger than in 7(h). The patterns in the $\delta\tilde{\rho}'$-$\delta\mathbf{b}'$ covariances, Figs. 7(c) and (i), have similar patterns but are larger in (c) for similar reasons.

For each experiment, the initial background is found by perturbing the initial truth in a way consistent with the $\mathbf{B}$-matrix of each experiment (see last bullet point in Sect. 4.6). All other aspects of the assimilation are identical in all three experiments: 3DFGAT with a one-hour window is used for 30 DA cycles. The observations are identical in all cases, which are generated from the truth run with an error standard deviation of 0.0015. There are 2520 observations of $\tilde{\rho}'$ in each cycle, which are spread over a $20 \times 18$ grid of points broadly covering the domain, and over seven times ($t = 0, 600, 1200, 1800, 2400, 3000, 3600$ s). Only one outer loop of 3DFGAT is executed per cycle with 100 inner loop iterations. This comparison is intended to be only a brief demonstration of the ABC-DA system, and we leave a more in-depth study to later papers.

## 5.2 Results for experiment GB+VR+ (control experiment)

Figure 9 shows how the cost function (panel a) and the size of its gradient (b) reduce with inner loop iteration of the conjugate gradient procedure for one cycle of the GB+VR+ experiment. The total cost function, $J$ (black in a) decreases along with the observation part, $J^o$ (red in a) and the gradient (b), at the expense of the background part of the cost function $J^b$ (blue in a), as expected. These results appear to show that the cost function has minimized. Figure 10 shows histograms of a selection of differences in the $\tilde{\rho}'$ observation space (for the same cycle as Fig. 9). The top row concerns differences between the observations and $\mathcal{H}(\mathbf{x}^b)$ (a, where $\mathcal{H}(\bullet)$ is the combined ABC model and observation operator for the observation network described at the end of Sect. 5.1), $\mathcal{H}(\mathbf{x}^a)$ (b), and $\mathcal{H}(\mathbf{x}^t)$ (c). The width (standard deviation) of each is shown in the heading of each panel, showing a slight reduction from (a) to (b) as expected for the DA to have worked. The analysis is closer to the truth than is the background as expected as revealed with the histograms on the bottom row, which are $\mathcal{H}(\mathbf{x}^b) - \mathcal{H}(\mathbf{x}^t)$ (d) and $\mathcal{H}(\mathbf{x}^a) - \mathcal{H}(\mathbf{x}^t)$ (e).

Diagnostics collected over all 30 cycles for each variable in this experiment are shown in Fig. 11. The left panels show how the root-mean-squared (RMS) values of the fields vary in time for three versions of the fields: the truth (solid lines), the analysed fields (dotted), and a free forecast starting from the original background (dashed). For all fields except $w$, the assimilation overestimates the RMS field values, although the assimilation does often follow the oscillations in the true trajectory. The overestimation in the assimilation is actually nearly always worse than for the free background for $v$, is sometimes worse,

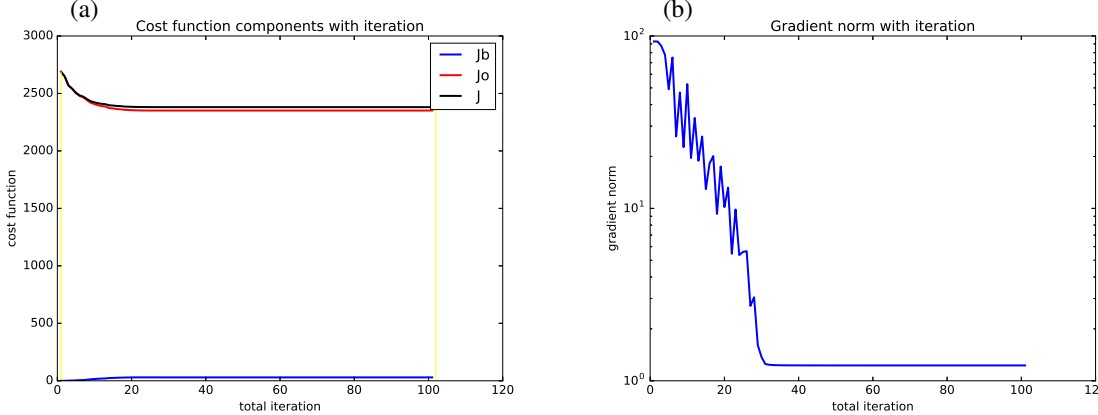

**Figure 9.** Cost function statistics as a function of inner loop iteration from the seventh cycle of the GB+VR+ experiment. Panel (a) shows the cost function contributions: $J^b$ (blue), $J^o$ (red), and $J = J^b + J^o$ (black). The yellow lines mark the zeroth and hundredth (the maximum iteration) iterations. Panel (b) shows the norm of the cost function gradient.

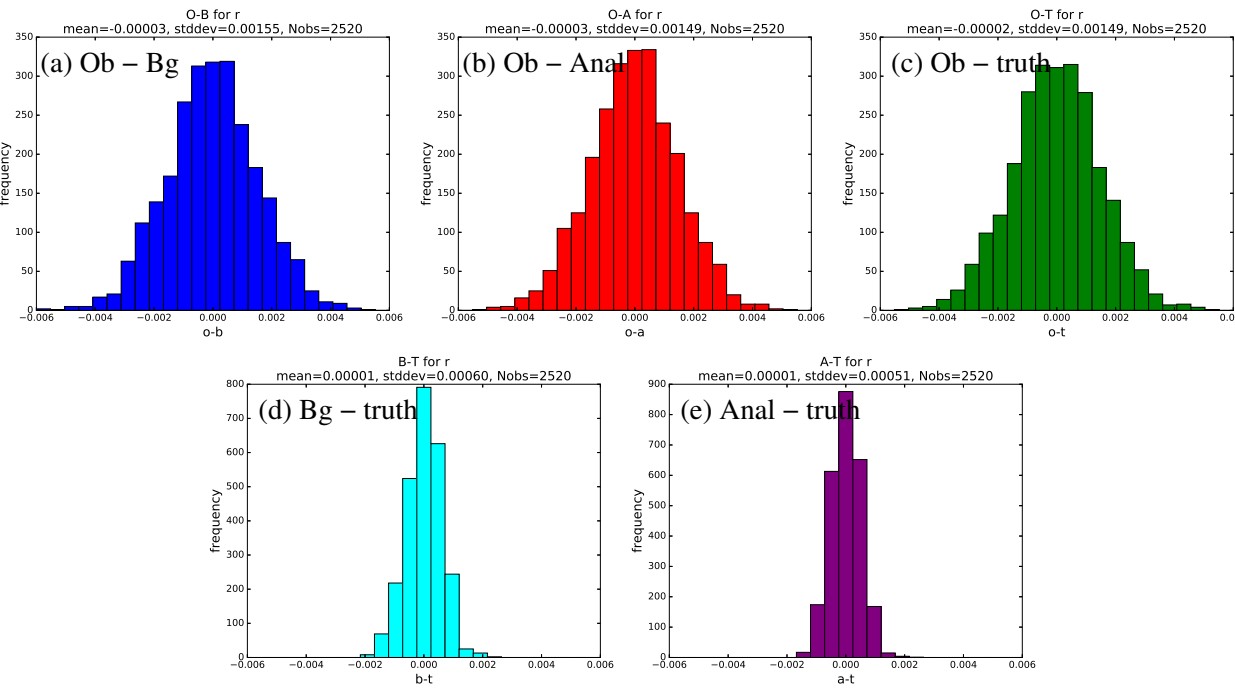

**Figure 10.** Panels (a), (b), and (c): frequency histograms of $y - \mathcal{H}(\mathbf{x})$ for the seventh cycle of the GB+VR+ experiment, where $\mathbf{x}$ is (a) $\mathbf{x}^b$ and (b) $\mathbf{x}^a$, and (c) $\mathbf{x}^t$. Panels (d) and (e): histograms of $\mathcal{H}(\mathbf{x}) - \mathcal{H}(\mathbf{x}^t)$ where $\mathbf{x}$ is (d) $\mathbf{x}^b$ and (e) $\mathbf{x}^a$. Observations are those assimilated of $\tilde{\boldsymbol{\rho}}'$.

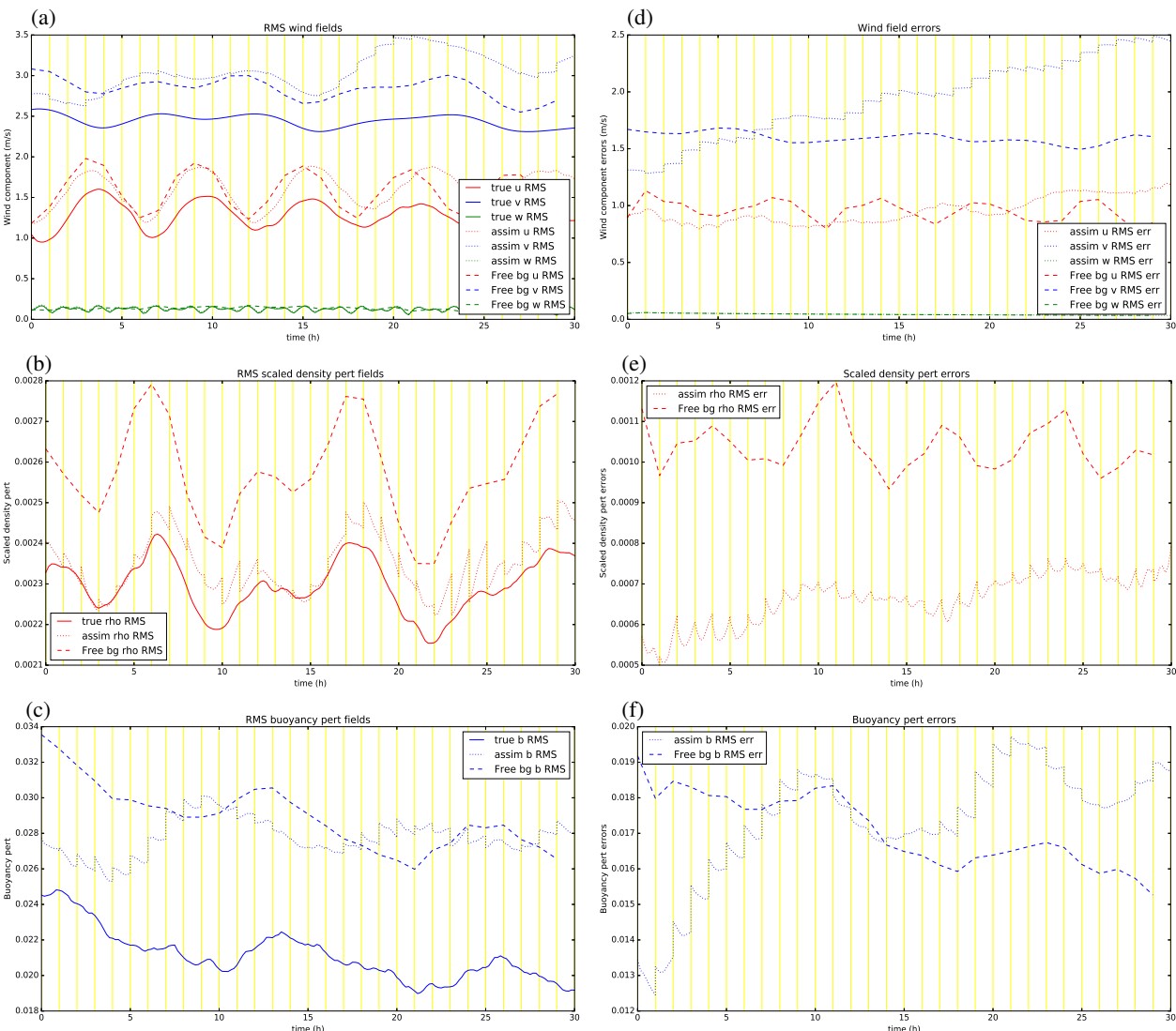

**Figure 11.** Left panels: RMS fields of (a) $u$, $v$, $w$, (b) $\tilde{\rho}$' and (c) $b'$ for the control experiment, GB+VR+. The solid lines are for the truth trajectory, the dotted lines are for the assimilation trajectory, and the dashed lines are for the free trajectories starting from the initial background. The vertical yellow lines mark the times of the cycle boundaries where the assimilation trajectory is the analysis at the start of each cycle and the background (for the next cycle) at the end. Right panels: RMS values of the errors (trajectory minus truth) for the corresponding quantities in the left panels.

sometimes better for $u$ and $b'$, but nearly always better for $\tilde{\rho}'$ (recall that only observations of $\tilde{\rho}'$ are assimilated). The right panels show the RMS errors (RMSE) of the analysed (dotted) and free (dashed) fields, which we do know in such twin experiments. In the first few cycles the analyses are improved compared to the free run in all variables, but there comes a time where the assimilation becomes worse than the free run for most fields (especially $v$ and $b'$). The assimilation stays better than the free run for $\tilde{\rho}'$ over all 30 cycles, and the background error covariances imposed between $\tilde{\rho}'$-$v$ and $\tilde{\rho}'$-$b'$ (panels b and c of Fig. 7) do not appear to be of benefit to $v$ and $b'$, which is surprising. Note that there are no assimilation jumps in the $u$ field because it is neither an observed variable nor correlated to $\tilde{\rho}'$ in the $\mathbf{B}$-matrix.

At $t = 0$ each two trajectories of the same colour in each right panel represent background (dashed) and analysis (dotted) errors and so can be compared. For all variables (except $u$ and $w$), these two points are different at $t = 0$, with smaller analysis errors than background errors. This indicates that the DA does add improvement early in the cycling. Even though the analyses are closer to the observations than the backgrounds, the errors in the assimilated variables do grow with time and most of the analysis increments (the jumps at the yellow cycle boundaries) do act to increase the RMS error throughout the experiment. There are a number of possible reasons for this degradation, e.g. a poor $\mathbf{B}$-matrix formulation for this model (including the use of balance constraints in the CVT), an inappropriate population of forecasts used to calibrate the $\mathbf{B}$-matrix, the use of 3DFGAT rather than 4DVar, or the particular choice of observation network for this convective-scale system. Given that the assimilation performs well at $t = 0$ (where the error in the initial background is consistent with the $\mathbf{B}$-matrix formulation by design), it could be that, as the cycling progresses, the nature of actual background errors become very different to those assumed by the $\mathbf{B}$-matrix. Confirming this hypothesis would require recalibrating the $\mathbf{B}$-matrix, which is outside the scope of the present paper. We focus now on the remaining two experiments described in Sect. 5.1 to see if any changes can be made by adjusting parts of $\mathbf{B}$.

### 5.3    Results for experiment GB-

Diagnostics for the GB- experiment are shown in Fig. 12. Comparing the RMS values (left) with the GB+VR+ experiment, the analysis RMS values become further from the truth for $u$, $v$, and $b'$. Note that, although the true trajectories are exactly the same as those of GB+VR+, the free trajectories are different as the initial background is consistent with a different $\mathbf{B}$-matrix. Again comparing to GB+VR+ the RMSE analysis errors (right panels) are generally worse for GB-, but the assimilation is still able to control the $\tilde{\rho}'$ field. This suggests that the $\mathbf{B}$-matrix for GB- is still not consistent with errors in the one-hour background forecasts over the cycles. Note that there are no assimilation jumps in the $u$ nor $v$ fields because they are not correlated to $\tilde{\rho}'$ in the $\mathbf{B}$-matrix.

### 5.4    Results for experiment GB+VR-

VR is designed to accompany the GB operator and the configuration that included these two together in the CVT may have contributed to the relatively poor performance in Sect. 5.2. Here we run the experiment that includes the GB step in the CVT, but not the VR. Diagnostics for the GB+VR- experiment are shown in Fig. 13. Again, the true state is the same as in the previous two experiments, but the free background forecast is different due to the different $\mathbf{B}$-matrix from which the initial

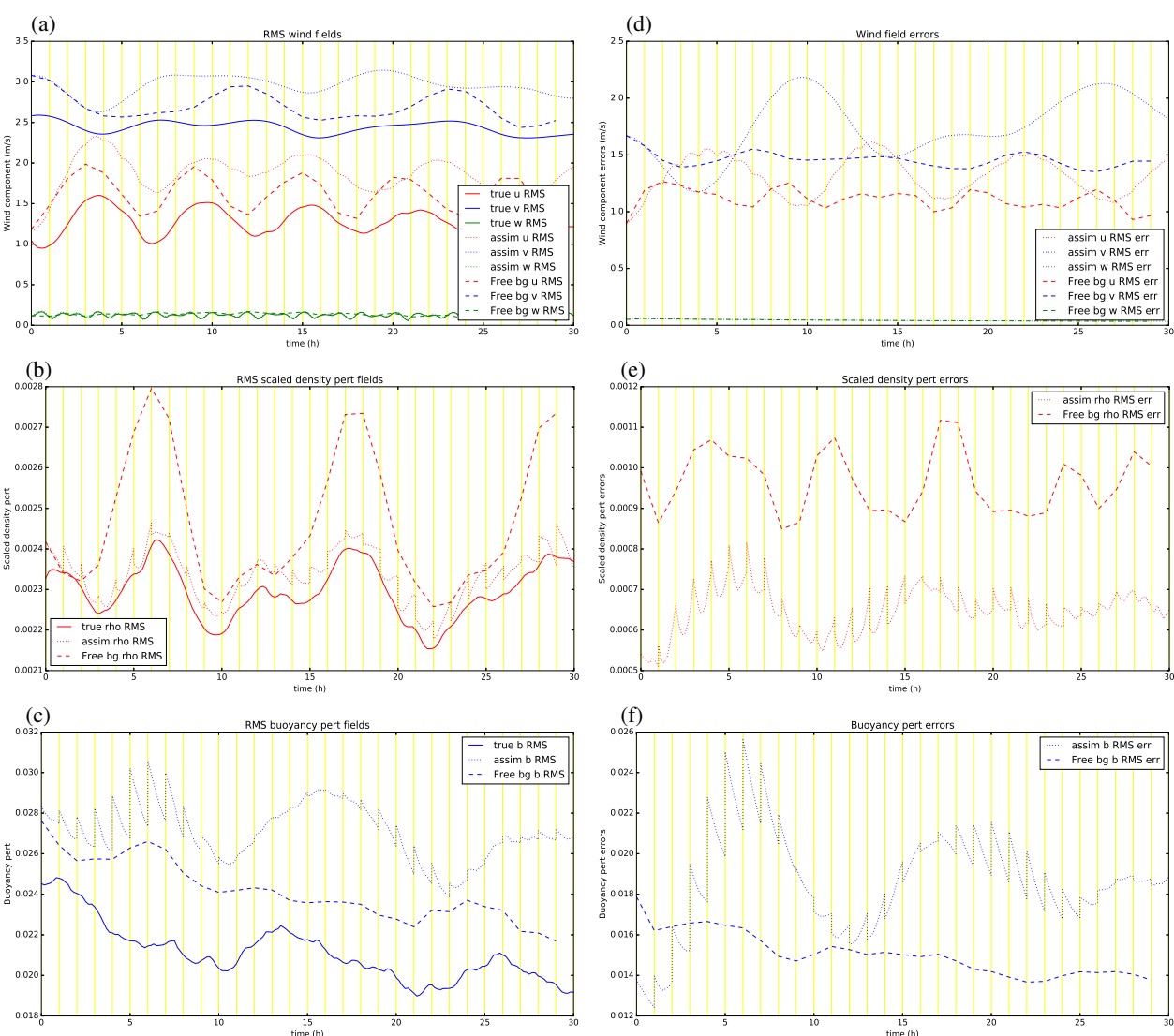

**Figure 12.** As Fig. 11, but for experiment GB-.

background state is sampled from. The RMS values (left panels) of the assimilation are generally closer to the truth than in GB+VR+ (apart from $u$ where they are about the same). The RMSE values (right panels) are also smaller as a fraction of the error in the free run (or at least are smaller for a longer time), although the errors in $v$ do still eventually fail to be controlled by the DA.

The absence of VR does have the effect of weakening the effective strength of the GB between $\delta\tilde{\rho}'$ and $\delta v$ – compare panels b and h of Fig. 7. This is because, overall, $\mathbf{R}_\rho$ enhances the balanced scaled density (and so the balanced scaled density is diminished when removing it). It is possible then that it is not the adoption of GB that results in the relatively poor performance of the GB+VR+ assimilation in this system, but the use of the VR.

## 6   Summary

This article is a documentation of ABC-DA, a Var system for use with the convective-scale ABC model (Petrie et al., 2017) to efficiently test new ideas for convective-scale DA. The DA currently has incremental 3DVar and 3DFGAT implemented although expansion to 4DVar, ensemble-variational, and hybrid methods is possible. The DA system has an expandable observation operator, which is currently configured to assimilate observations of all model variables, and 2D (horizontal) and 3D wind speeds. The choice of observation network is not fixed, and it is possible to specify arbitrary observation networks in space and time. The $\mathbf{B}$-matrix is modelled in a very compact form by a control variable transform (CVT), which is implemented with a range of options. The current multivariate options include the option to use geostrophic balance (GB) to couple $\delta\tilde{\rho}'$ and $\delta v$, hydrostatic balance (HB) to couple $\delta\tilde{\rho}'$ and $\delta b'$, and anelastic balance to couple $\delta u$ and $\delta w$. All of these balances use diagnostic relationships using analytical (rather than statistical) operators. A vertical regression (VR) step is also included (which works with the geostrophic step), to mirror the operational set-up of the Met Office's DA system. While the model increments are $\delta u$, $\delta v$, $\delta w$, $\delta\tilde{\rho}'$, and $\delta b'$, the control parameters are $\delta\psi$, $\delta\chi$, $\delta\tilde{\rho}'^{\mathrm{u}}$, $\delta b^{\mathrm{u}}$, and $\delta w^{\mathrm{u}}$, which are assumed to be mutually uncorrelated. The transforms that model the spatial covariances for each control parameter are formed of vertical and horizontal parts, which model covariances in these respective directions. There are further options that control how each of these spatial transforms work, including the order in which they are used.

ABC-DA comprises component suites to perform the following tasks.

– Run a pure model forecast of any length starting from specified initial conditions.

– Investigate linear properties of the equations of motion.

– Generate an initial ensemble of states (e.g. for calibration of the $\mathbf{B}$-matrix).

– Calibrate the parameters that describe components of the CVT to produce a CVT file. This specifies the $\mathbf{B}$-matrix for use in the DA program.

– Test the components of the system (e.g. adjoint tests of CVT operators).

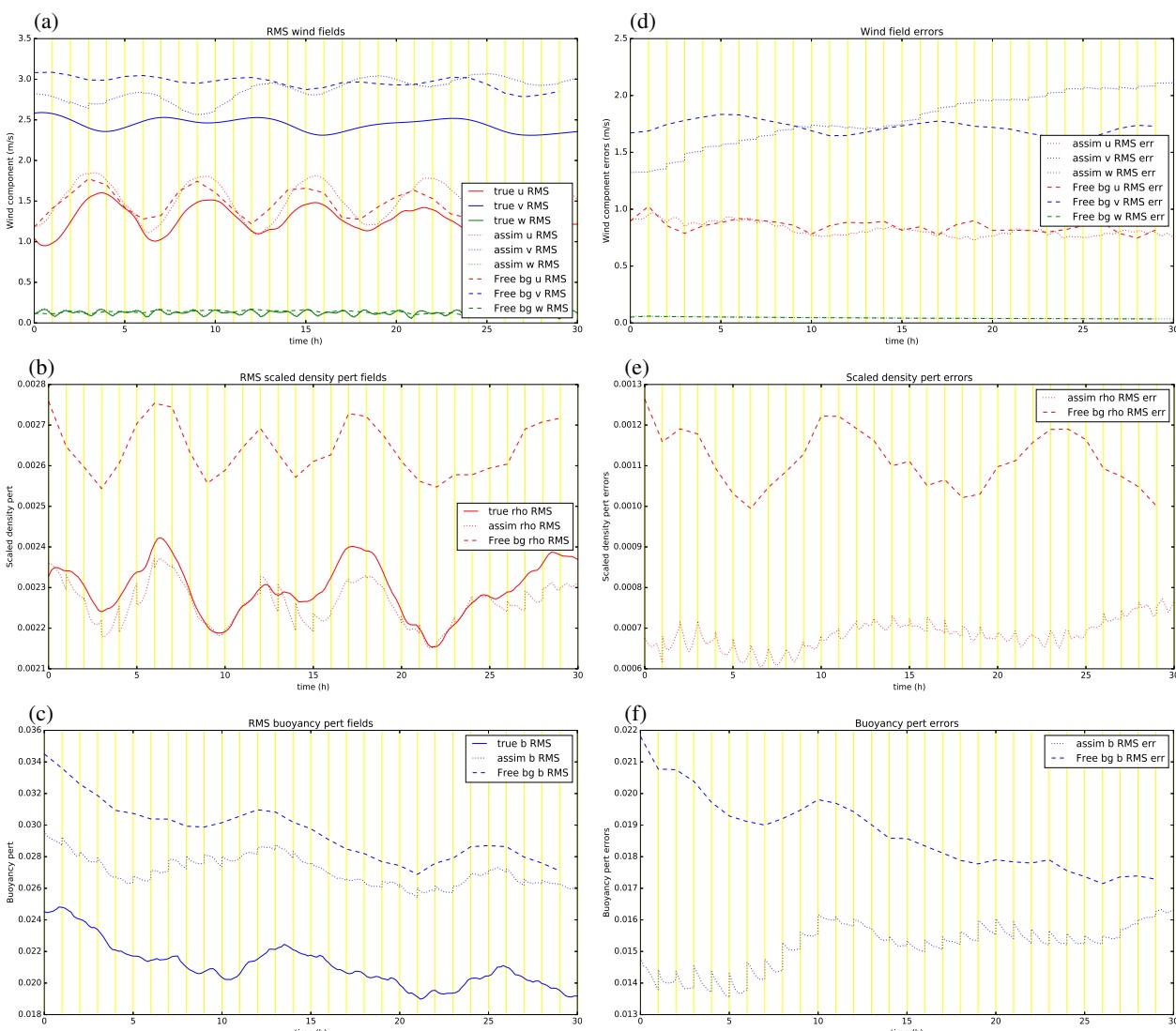

**Figure 13.** As Fig. 11, but for experiment GB+VR-.

- Generate 'raw' covariance maps using a specified population of states.

- Generate implied covariance maps from a specified CVT file.

- Generate sample observations (consistent with a specified $\mathbf{R}$-matrix and truth) and a background state (consistent with a specified $\mathbf{B}$-matrix and truth).

- Perform a cycle of Var with a specified background state, observations, and CVT file.

- Perform a run of cycled DA/forecast steps where the analysis of one cycle is used to make a background forecast for the next.

The number-crunching code is written in Fortran 90/95, the scripts are written in Bash, and the example plotting routines are written in Python 2.

A selection of diagnostics of the CVT is provided in this paper and a brief study is made to illustrate the system's performance when GB (and VR) is switched on and off in the CVT. GB is used to model the background error coupling between $\delta\tilde{\rho}'$ and $\delta v$, which is zero when GB is not used, and has a classical dipole pattern ($\delta v$ response to a point perturbation in $\delta\tilde{\rho}'$) when it is used. Exploiting this balance constraint (or not) is investigated as it is a question whether it has validity in the description of convective-scale background error covariances. VR has the effect of strengthening this geostrophic relationship because it increases (decreases) the variance of the balanced (unbalanced) contributions to $\delta\tilde{\rho}'$.

Experiments are performed using a set of three 3DFGAT twin experiments by assimilating observations over a 30 hour forecast taken as the common truth. The three experiments are (i) using GB and VR (GB+VR+), (ii) not using GB (GB-), and (iii) using GB but not VR (GB+VR-). Each experiment has a different CVT (i.e. a different $\mathbf{U}$) and hence a different implied background error covariance matrix, $\mathbf{B}^{\mathrm{ic}} = \mathbf{U}\mathbf{U}^{\mathrm{T}}$. An initial background state is constructed for each experiment by perturbing the truth at $t = 0$ with $\mathbf{U}\delta\chi$ where $\delta\chi$ is a sample from $N(0,\mathbf{I})$ (equivalent in model space as choosing a sample from $N(0,\mathbf{B}^{\mathrm{ic}})$). The DA/forecast is cycled hourly over 30 cycles with 2520 observations of $\tilde{\rho}'$ per cycle. The results are summarised as follows.

- The GB+VR+ experiment does not result in a systematic reduction of errors in the model variables. In fact, for some variables (namely $v$ and $b'$), it leads to an increase in error of the analysis above the error level of the free background.

- The GB- experiment represents a degradation in performance of the unobserved variables in that the errors are above the error level of the free background for more time than in GB+VR+.

- The GB+VR- experiment represents the configuration that is the best of those tested. The errors in all variables except $v$ are kept below the error level of the free background.

It is surprising that errors in one of the variables in particular ($v$) cannot be properly corrected in these experiments. It is postulated that this outcome is not due to an inappropriate $\mathbf{B}$-matrix (for instance in Figs. 11 and 13), but due to the use of 3DFGAT. Separate tests with 3DVar and with all observations made at the analysis times, rather than spread throughout the

analysis window (not shown), suggest that observations of $\tilde{\rho}'$ can reduce the error in $v$. We have learned from these experiments that nature of the **B**-matrix can have a significant impact on the performance of the DA. Application of GB in such a convective-scale system has a complicated effect: while experiment GB+VR- is better than experiment GB- overall, the application of GB does systematically increase errors in the $v$ field over the 30 cycles, and 'too much' GB (as in experiment GB+VR+) enhances this effect. Similar studies may be done by turning on and off hydrostatic and anelastic balances.

These results highlight a difficulty of the convective-scale DA problem and they raise more questions about how this system could be improved and how it could contribute to advances in DA. A possible way forward might be to keep the current formulation of the CVT but to calibrate it with data from a different source, either as an ensemble, as differences between pairs of forecasts valid at the same time (the popular National Meteorological Center method (Parrish and Derber, 1992; Berre et al., 2006)), or from a single long forecast (the so-called 'Canadian quick' method, Polavarapu et al., 2005). Other ways forward might be to replace the traditional geophysical balances with a new balance relationship that may be more suitable at convective-scales, such as the diagnostic equation for non-hydrostatic pressure outlined in Sect. 4.3 of Pielke (2002). The CVT may also though be adapted so that the DA system acts as an ensemble-variational (EnVar) system where the analysis increments are expressed as linear combinations of ensemble perturbations (e.g. Lorenc (2003); Buehner (2005)), or as a hybrid-EnVar system (a hybrid of the existing **B**-matrix and the EnVar, e.g. Wang et al. (2007)), or as part of a hybrid gain formulation (e.g. Penny (2014)). These approaches imply flow-dependent background error covariances (e.g. Bannister, 2017), but each would require a separate ensemble DA system parallel to the variational one, and would also require localisation techniques to mitigate for rank deficiency of the ensemble. These are entirely possible developments given appropriate resources. Studying the **B**-matrix is the main application that the author has in mind, although the system could equally well be used to study impacts of new high-resolution observations.

*Code and data availability.* The model and data assimilation system are written in Fortran-90/95, and the plotting code is written in python. This software is open-source and freely available on a Git Hub repository (Bannister, 2019), DOI: 10.5281/zenodo.3531926. A sample of initial condition data is also freely available, DOI: 10.5281/zenodo.3946359.

*Competing interests.* The author declares that he has no conflicts of interest.

### Acknowledgements

The author is supported by the National Centre for Earth Observation (NCEO), which is funded by the Natural Environment Research Council (NERC). The author acknowledges use of the following open source libraries: netCDF, FFTpack, LApack, tmglib, refblas, and matplotlib. The author is grateful to two anonymous referees and Prof. Pierre Gauthier for constructive comments on a previous version of this paper.

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

**Appendix A: Number of pieces of information to make up a covariance model**

This appendix shows the number of pieces of information that specifies the covariance model used in this paper. Since only analytic balance relationships are used (rather than statistically-derived ones, which would themselves require calibration
from data), components only of the spatial transform ($\mathbf{U}_s$) and the vertical regression ($\mathbf{R}_\rho$) require calibration. Referring to Eq. (31) or Eq. (32), this includes determination of the standard deviation matrix ($\mathbf{\Sigma}$), and components of the vertical ($\mathbf{U}_v = \mathbf{F}_v \mathbf{\Lambda}_v^{1/2}$) and horizontal ($\mathbf{U}_h = \mathbf{F}_h \mathbf{\Lambda}_h^{1/2}$) trasforms. The vertical regression, $\mathbf{R}_\rho = \mathbf{C}^{\delta\tilde{\rho}'\delta\tilde{\rho}'^b} \left( \mathbf{C}^{\delta\tilde{\rho}'^b \delta\tilde{\rho}'^b} \right)^{-1}$ requires

| Matrix | Description | Elements to be determined |
|---|---|---|
| $\mathbf{\Sigma}$ | Standard deviations (level-dependent only) | 60 |
| $\mathbf{F}_\mathrm{v}$ | Eigenvectors of vertical transforms | $60 \times 60$ |
| $\mathbf{\Lambda}_\mathrm{v}^{1/2}$ | Eigenvalues of vertical transforms | 60 |
| $\mathbf{F}_\mathrm{h}$ | Horizontal Fourier transform | 0 |
| $\mathbf{\Lambda}_\mathrm{h}^{1/2}$ | Horizontal variances of each parameter vertical mode | $180 \times 60$ |
| | Total per control parameter (sum of above) | 14 520 |
| | For all five control parameters | 72 600 |
| $\mathbf{C}^{\delta\tilde{\rho}'\delta\tilde{\rho}'^\mathrm{b}}$ | Non-symmetric term in $\mathbf{R}_\rho$ | $60 \times 60$ |
| $\mathbf{C}^{\delta\tilde{\rho}'^\mathrm{b}\delta\tilde{\rho}'^\mathrm{b}}$ | Symmetric term in $\mathbf{R}_\rho$ | $60 \times 61/2$ |
| | Grand total | 78 030 |

**Table A1.** Number of components of the spatial transforms that need to be estimated from data. The number of elements in column 3 are for the standard ABC system of 360 longitudes and 60 levels. Note that for $\mathbf{\Lambda}_\mathrm{h}^{1/2}$, 180 wavenumbers are specified rather than 360 as the negative wavenumbers are assumed to have the same variances as the positive ones.

determination of a non-symmetric and symmetric correlation matrix. Table A1 shows this information, concluding that 78 030 pieces of information are needed. In this paper, a super-ensemble of 260 members is used to estimate this background error covariance model. This provides $260 \times 5 \times 360 \times 60 = 28\,080\,000$ pieces of information, vastly more than the number of elements needed for the covariance model. Compare this to $n(1+n)/2$ elements to determine a full background error covariance matrix, where $n = 5 \times 360 \times 60$ is the number of elements in the model state vector. This evaluates to 5 832 054 000, requiring 54 000 ensemble members to estimate these elements at the very least. While this is possible, it is not practical at present.

## Appendix B: Vertical regression of balanced scaled density

The direct calculation of the geostrophicaly balanced scaled density from Eq. (2a) (in practice found via the stream-function as in Eq. (15)) leads to difficulties with regard to the vertical continuity of the resulting $\delta\tilde{\rho}'^\mathrm{b}(x,z,t)$ (Lorenc et al., 2000). If a function (arbitrarily chosen but independent of $x$, i.e. $p(z,t)$) is added to $\delta\tilde{\rho}'^\mathrm{b}$, then $\delta\tilde{\rho}'^\mathrm{b} + p$ is also a solution of Eq. (2a). This will potentially allow damaging jumps to appear with short vertical scale. According to geostrophic adjustment theory, stream-function (and hence the scaled density computed from it) is not balanced for short vertical scales (e.g. Cullen (2003)) and so the output from Eq. (15) is smoothed in the vertical. This is done by modifying the balanced scaled density with a regression operator $\mathbf{R}_\rho$.

From hereon in this appendix, we drop the $\boldsymbol{\rho}$ subscript on $\mathbf{R}_\rho$ for brevity as the subscript will be used to indicate matrix element. We also for brevity drop the tilde, prime, and $\delta$ notations on the scaled density fields, remembering that the fields are implicitly perturbations. One strategy for $\mathbf{R}$ is to introduce similar vertical length-scales as those in the forecasts of total scaled

density, $\delta\rho$. We therefore design $\mathbf{R}$ to minimise the following penalty:

$$\mathbf{R} = \arg\min_{\mathbf{R}} \mathcal{J}[\mathbf{R}] = \frac{1}{2}\frac{1}{N}\sum_{i=1}^{N}\left(\mathbf{R}\boldsymbol{\rho}^{\mathrm{b}}(i) - \boldsymbol{\rho}(i)\right)^{\mathrm{T}}\left(\mathbf{R}\boldsymbol{\rho}^{\mathrm{b}}(i) - \boldsymbol{\rho}(i)\right), \tag{B1}$$

where $\boldsymbol{\rho}^{\mathrm{b}}(i)$ and $\boldsymbol{\rho}(i)$ are the $i$th member of an $N$-member population of balanced and total scaled density perturbations respectively. To find the $\mathbf{R}$ that minimises $\mathcal{J}$, we solve the equation $\partial\mathcal{J}/\partial\mathbf{R} = 0$, which is a matrix equation, with element $a, b$ being $\partial\mathcal{J}/\partial\mathbf{R}_{ab} = 0$. It is convenient to expand the elements of Eq. (B1):

$$\mathcal{J}[\mathbf{R}] = \frac{1}{2}\frac{1}{N}\sum_{i=1}^{N}\sum_{\alpha}\left(\sum_{\beta}\mathbf{R}_{\alpha\beta}\boldsymbol{\rho}_{\beta}^{\mathrm{b}}(i) - \boldsymbol{\rho}_{\alpha}(i)\right)^{2}, \tag{B2}$$

where all subscripts indicate matrix or vector element. Differentiating with respect to $\mathbf{R}_{ab}$ gives:

$$\begin{aligned}\frac{\partial\mathcal{J}}{\mathbf{R}_{ab}} &= \frac{1}{N}\sum_{i=1}^{N}\sum_{\alpha}\left(\sum_{\beta}\mathbf{R}_{\alpha\beta}\boldsymbol{\rho}_{\beta}^{\mathrm{b}}(i) - \boldsymbol{\rho}_{\alpha}(i)\right)\frac{\partial}{\mathbf{R}_{ab}}\left(\sum_{\beta'}\mathbf{R}_{\alpha\beta'}\boldsymbol{\rho}_{\beta'}^{\mathrm{b}}(i)\right), \\ &= \frac{1}{N}\sum_{i=1}^{N}\sum_{\alpha}\left(\sum_{\beta}\mathbf{R}_{\alpha\beta}\boldsymbol{\rho}_{\beta}^{\mathrm{b}}(i) - \boldsymbol{\rho}_{\alpha}(i)\right)\sum_{\beta'}\delta_{a\alpha}\delta_{b\beta'}\boldsymbol{\rho}_{\beta'}^{\mathrm{b}}(i), \\ &= \frac{1}{N}\sum_{i=1}^{N}\left(\sum_{\beta}\mathbf{R}_{a\beta}\boldsymbol{\rho}_{\beta}^{\mathrm{b}}(i) - \boldsymbol{\rho}_{a}(i)\right)\boldsymbol{\rho}_{b}^{\mathrm{b}}(i), \end{aligned} \tag{B3}$$

where $\delta_{\alpha\beta}$ in the second line is the Kroneker delta-function. Now this result is put back into matrix notation. Note that $\left(\sum_{\beta}\mathbf{R}_{a\beta}\boldsymbol{\rho}_{\beta}^{\mathrm{b}}(i) - \boldsymbol{\rho}_{a}(i)\right)\boldsymbol{\rho}_{b}^{\mathrm{b}}(i)$ is the $a, b$th element of the outer product matrix $\left(\mathbf{R}\boldsymbol{\rho}^{\mathrm{b}}(i) - \boldsymbol{\rho}(i)\right)\left(\boldsymbol{\rho}^{\mathrm{b}}(i)\right)^{\mathrm{T}}$, so Eq. (B3) is:

$$\frac{\partial\mathcal{J}}{\mathbf{R}} = \frac{1}{N}\sum_{i=1}^{N}\left(\mathbf{R}\boldsymbol{\rho}^{\mathrm{b}}(i) - \boldsymbol{\rho}(i)\right)\left(\boldsymbol{\rho}^{\mathrm{b}}(i)\right)^{\mathrm{T}}.$$

Setting this to zero to solve for the optimal regression matrix gives:

$$\mathbf{R}\frac{1}{N}\sum_{i=1}^{N}\left(\boldsymbol{\rho}^{\mathrm{b}}(i)\right)\left(\boldsymbol{\rho}^{\mathrm{b}}(i)\right)^{\mathrm{T}} = \frac{1}{N}\sum_{i=1}^{N}\left(\boldsymbol{\rho}(i)\right)\left(\boldsymbol{\rho}^{\mathrm{b}}(i)\right)^{\mathrm{T}}.$$

The outer product matrix – summed and divided by $N$ – on the left-hand-side is identified as the covariance matrix of the balanced scaled density perturbations with themselves ($\mathbf{C}^{\boldsymbol{\rho}^{\mathrm{b}}\boldsymbol{\rho}^{\mathrm{b}}}$), and the corresponding terms on the right-hand-side is the covariance matrix between the total scaled density perturbations and the balanced scaled density perturbations ($\mathbf{C}^{\boldsymbol{\rho}\boldsymbol{\rho}^{\mathrm{b}}}$). The optimal regression matrix is then

$$\mathbf{R} = \mathbf{C}^{\boldsymbol{\rho}\boldsymbol{\rho}^{\mathrm{b}}}\left(\mathbf{C}^{\boldsymbol{\rho}^{\mathrm{b}}\boldsymbol{\rho}^{\mathrm{b}}}\right)^{-1}. \tag{B4}$$

## Appendix C: Estimation of characteristic length-scales

There is a well-defined length-scale associated with a plane wave of the form $\exp i\kappa x$ ($\kappa$ is wavenumber, $x$ is position), this being the wavelength, $l = 2\pi/\kappa$. For a periodic domain of length $L$, the allowed wavenumbers are $\kappa = 2k\pi/L$ ($k$ wavenumber

index, integer), and so the wavelength is $L/k$. There is no similar analytical relationship between the vertical mode index, $\nu$, and the vertical length-scale, $l_{\mathrm{v}}$, but it is possible to estimate the characteristic length-scale as follows. Suppose that $f(z)$ is an oscillatory function (e.g. a particular vertical eigenvector (column of $\mathbf{F}_{\mathrm{v}}$) of one of the vertical covariance matrices introduced in Sect. 4.2.3). Let its Fourier transform be $\bar{f}(m)$, where $m$ is the spectral co-ordinate (i.e. $f(z) \propto \sum_m \bar{f}(m)\exp i2m\pi z/L_{\mathrm{v}}$, where $L_{\mathrm{v}}$ is the vertical extent of the model). Treating $\left|\bar{f}(m)\right|^2$ as the weight of the contribution from wavenumber index $m$, the characteristic value of $m$ is:

$$\langle m \rangle = \sum_m \left|\bar{f}(m)\right|^2 m \Big/ \sum_m \left|\bar{f}(m)\right|^2 . \tag{C1}$$

This translates to a vertical length-scale of $l_{\mathrm{v}} = L_{\mathrm{v}}/\langle m \rangle$. This is the procedure used to make Fig. 6b. We note that the Fourier transform assumes periodicity, which is not a property of the fields in the vertical direction, but since we are after only a rough estimate of $l_{\mathrm{v}}$, we assume that this inconsistency does not have a major impact.