# Peer review of "The "ABC-DA system" (v1.4): a variational data assimilation system for convective scale assimilation research with a study of the impact of a balance constraint"

_Geoscientific Model Development, 2019_

## Referee Comment (RC1) · Anonymous Referee #1 · 30 Apr 2020

This paper describes a DA system that has been built for the ABC model of Petrie et al. (2017). The entire system, ABC-DA, is described as flexible, configurable, and efficient enough to be run on a personal computer. One of the stated applications of the ABC-DA is to study convective-scale DA. This system could also clearly be used as a teaching aid for DA students. Section 4 gives a tutorial on variation data assimilaton and a practical "how to" for constructing a 3DVar system. Section 4 gives a tutorial on control variable transforms (CVTs) and is also a practical "how to" for developing CVTs. These two sections are well written tutorials which, by themselves, could serve as a
good teaching aids.

I would recommend the publication of this paper after the author addresses a few minor comments.

1. Figure 7 might be more easily interpreted if the correlation rather covariance were plotted. It is not easy evaluate the importance of the cross-covariances (columns 2 & 3).

2. A ensemble of forecast perturbations is used for training/developing the CVT. The author describes this raw ensemble in this way: "We regard the raw covariances as a guide to the 'true' covariances that should ideally be modeled by the CVT." In principal, the ensemble could be made large enough to provide a very accurate covariance, from which the implied covariances (from CVT choices) could be directly evaluated. Is the model state size too large to create a full rank (or nearly full) for this type of comparison?

3. Following on comment #2. If the training ensemble is of low rank, then it is well known that the covariances must be localized. It is possible that the covariance used for the experiments (i.e. bottom row of Figure 7) should be localized before used for developing the CVT?

4. There is much current research on using ensembles to represent the B matrix in variational DA. Can you comment on any plans to incorporate the ability to directly use an ensemble to perform the background covariance multiply, or possibility a hybrid approach?

GMDD

---

## Referee Comment (RC2) · Anonymous Referee #2 · 3 May 2020

Review of the GMD submitted article referenced as:

Title: The "ABC-DA system" (v1.4): a variational data assimilation system for convective scale assimilation research with a study of the impact of a balance constraint Author(s): Ross Noel Bannister MS No.: gmd-2019-318 MS Type: Model description paper Iteration: Minor Revision

The paper presents a very comprehensive technical description of a toy data assimilation system based on the previously published ABC-model formulation and codes. Its

content is extremely detailed and can easily serve as useful introduction to any scientist, including young scientists and post-graduate students, interested in uploading and using the codes for research or education. The technical content is complemented by a showcase example of the scientific use of the system (the study of the impact of the ABC-associated geostrophic balance constraint in the B-matrix model).

The article is clearly written and the figures are of good quality. The presentation as a whole matches the expected goal which is to provide a scientific introduction to the ABC-DA system. As a reviewer of this article, I do not consider that my role is to evaluate or critizise whether the ABC-model and DA formulation will be a useful scientific tool per se. The present paper will however enable the scientific data assimilation community evaluate that usefulness in practice. I therefore recommend the paper for publication after very minor revisions.

Hereafter follow my minor comments:

1) about §4.1: from what platform can a user upload UM data for initializing the very first steps ? Or are such data available with the ABC system packages ?

Ensembles play an important role in many cases of the toy applications. Can you say a few words about how the size of the super-ensemble is set, and what would be a "reasonable" limit of size ?

2) About Fig 6b: this particular plot is actually little discussed in the core text. My question is, noting that the vertical lengthscale increases with increasing vertical mode (i.e. the more nodes on the vertical, the deeper the penetration scale of the mode), is this behaviour due to the fact that the plot holds for the unbalanced part of the scaled density ? i.e. one expects the opposite property for the balanced part of scaled density (low-order vertical modes of balanced scaled density would have the largest vertical lengthscales ?). Is this correct ? I suggest a short comment about Fig 6b could be added in §4.3 or §4.4.

3) about §4.7: it is stated that the ABC-DA system is flexible enough to host a variety of DA methods, like 4D-VAR or Ensemble-Variational formulations. It seems indeed clear from the article that methods based on variational formulations, including iterative steps such as a minimisation and the computation of a gradient, are allowed. However, what about methods like Extended or Ensemble Kalman filters, or versions of Optimal Interpolation, i.e. methods where the Gain Matrix (G) would be somehow explicitly computed, and a direct inversion step involving G would be implied ? Similarly, what about methods involving a number of computational steps in observation space as for LETKF (Local Ensemble Transform Kalman filters) ? Can the author elaborate in only a few lines on these algorithms, in order to provide an insider view about how easy/how difficult/how different the implementation of such methods in ABC-DA would be ?

4) §6, line 776: typo "... that that ..." => "... that the ..."

5) line 774-780: in the discussion of the "control-ability" of the v-component of the wind fields. Is this weak control-ability due to the specific formulation of the ABC toy model ? (my guess is "yes"). Can you comment this more in the discussion ?

6) §4.2-4.3-4.4 & §5: One general question I have is whether the ABC-DA system can allow the use of a full, total field, B-matrix (that is, one without any balance modeling) ? If a total-field B-matrix would be feasible, then the corresponding ABC-DA system could be a valuable reference system for impact studies on specific B-modeling. Can you comment on this ?
* * *

---

## Short Comment (SC1) · 15 May 2020

Review of paper The ABC-DA system (version 1.4) : a variational data assimilation system for convective scale assimilation research with a study of the impact of a balance constraint By Ross Noël Bannister Submitted to Geoscientific Model Development Date:

1. MAIN COMMENTS The resolution of operational NWP models increases and is reaching the convective scale. This raises new issues associated with the inclusion of

ever increasing and complex processes. Moreover, this raises new questions regarding data assimilation methods to be used. Petrie et al. (2017) introduced a simple non-hydrostatic model that could even represent explicitly acoustic waves for data assimilation studies. This paper presents a companion variational data assimilation for this model. The paper is well written and focuses for the most part on the modeling of background error covariance. The emphasis is mostly on 3D-var including the so called FGAT (first-guess at appropriate time) which is also used in many operational systems.

In research, one would like to be able to explore different approaches to assess what could be the best one for the assimilation at the convective scale. I was expecting then that the 4D assimilation would have been more prominent, including 4D-Var in strong and weak constraint for example. In the EnVar, the control variable is the complete model trajectory including temporal correlations for the treatment of background error, which is also used in the weak constraint 4D-Var.

How does this approach compares to using the OOPS/JEDI paradigm which can be used with either a "toy " model or an operational one? I think that it would be important for a "community" model that the advantages of ABC-DA be presented from that perspective. Little is said about observation operators which can easily be the dominant component of a DA system. Assimilating large volumes of Doppler radar data is one example. The point I am making is the paper should make an effort to emphasize aspects of the ABC-DA that could entice researchers to use it.

As stated line 143, "much of the design of ABC-DA is concerned with how B is modelled". The emphasis is mostly on multivariate 3D-var including the so called FGAT (first-guess at appropriate time) which is also used in most operational system. Bouttier et al. (1997) introduced balance operators obtained from multilinear regression for a covariance model based on homogenous and isotropic correlations for the analysis variables deemed to have uncorrelated error. This corresponds to some extent to what the paper presents in too much details in my view. I do not see the point of explicitly

describing how to use the code: this is important but very likely to change with time. This should be included in a user manual or website for users. On the other hand, it would be more important to explain the scientific justifications. Balance appropriate for the convective scale should be discussed: geostrophic balance does not seem to be the most relevant.

The paper ends by presenting the results of experiments to illustrate the impact of the different components representing the balance for this particular model. Is the paper about presenting the ABC-DA emphasizing the advantages of the design to study different aspects of DA that may be important for the assimilation at the convective scale? As it is, most of it is to describe what has been implemented for this particular model of B with some results indicating the impact this may have on the analysis.

I recommend that the author reviews his paper to either present ABC-DA as a polyvalent system for research on DA at the convective scale. Or that it is about a multivariate model for B and its impact on the analysis and forecast. Given the large body of literature on this topic, the latter would be rather thin.

The next section presents some specific comments on some, but not all, issues with the paper.

2. SPECIFIC COMMENTS

2.1 Modelling B : sections 3.4, 4.2, 4.3, 4.4 (∼14 pages) As stated line 143, "much of the design of ABC-DA is concerned with how B is modelled". Sections The emphasis is mostly on 3D-var including the so called FGAT (first-guess at appropriate time) which is also used in most operational system. Early on, it has been recognized that multivariate covariances should embed dynamical constraints such as an approximate geostrophic balance (e.g., see Daley ,1991). Modeling of a "static" B has been the object of many papers that should be referred to Parrish and Derber (1992) presented the first implementation of 3D-Var and introduced a new approach in which the error was divided into balanced and unbalanced components. They used an ensemble of lagged

forecasts at 24 and 48-h to represent averaged background error covariances. Bouttier et al. (1997) introduced balance operators obtained from multilinear regression for a covariance model based on homogenous and isotropic correlations for the analysis variables deemed to have uncorrelated error. This is pretty much what is presented in section 4.2. I do not see why the author presents this with so much detail given that this is at best, an example of what could be used in the ABC-DA. Buehner (2005) presents a B based on a EOF representation for stationary covariances that can capture some local effects (e.g., presence of orography).

2.2 Other comments

p.7: section 3.3. devotes 7 lines to observations. Later, section 5 gives 8 more lines to the observation operators. This is a bit short in my view.

p.6, line 173: The propagator is said to be difficult to derive so it is replaced by the identity. It has been theoretically defined (LeDimet and Talagrand, 1988) and developed for operational model. Even more, the "transpose" of it has also been developed, the "adjoint model". In the context of the incremental form of 4D-Var, some simplifications to the model can be made regarding resolution or the used of a simplified physics. It would be important to know whether it should be possible to expand the ABC-DA to make it possible to do 4D-Var.

p.7, line 200: using the analysis variables $\delta(x - x\_b) = U$ with $B = UU^T$ means that we need to get the square root of B but we do not have to invert it. The flip side to this is that if B is singular the increment is built based on the singular vectors of B. For an ensemble like the ETKF, for instance, the increment could only be a linear combination of the members of the ensemble that define B. p.8, lines 207-213: I think this needs to be revised. What is said here only applies to a particular B model with isotropic and homogenous correlations which happens to yield a diagonal matrix when expressed in terms of spectral components (e.g., Fourier, Bessel or spherical harmonics).

p.8, line 224: in the incremental form, there is no need to invert U, insofar as the initial

point of the minimization is the background state, in which case, initially, \chi =0.

p.8, line 235: the description of the algorithm does not indicate how \lambda is updated.

p.9, line 245: I agree that the test of transpose is useful to test specific part of the code. But the gradient test (based on a Taylor expansion) should also be mentioned. It is simpler and is routinely used to validate complex operational variational DA systems when al components are active. It should be mentioned.

p.10, line 280: I do not think U-1 is needed.

p.10, line 290: the Helmholtz theorem states that there both a potential component and a rotational one (the streamfunction).

p.11, line 296: defining the balance operators is a separate exercise that needs more explanation. Using linear regression has been proposed by Bouttier et al. (1997) and used by others. This requires some insight into the type of balance that could exist. At synoptic scales, geostrophic and Ekman balance have been used to guide the linear regression. What kind of stationary balance can we expect at convective scales. Reference should be made to Parrish and Derber (1992) and Bouttier et al. (1997).

p.12, eq.(23): I think the winds cannot be represented as irrotational and this impacts the very formulation of U. Please revise. This impact the form of B as presented in eq.(22). p.14, line 366: I do not think we can represent B as a finite expansion of its eigenvectors, in general. It is a composition of operators that reflect the general form of the balance operators. In the univariate case for example, with homogeneous and isotropic correlations, this would require a large number of eigenvectors to properly represent it particularly if the characteristic scale is small.

p.15: section 4.3: one diagnostic used to calibrate the error statistics is to verify if the a priori statistics used in the assimilation are consistent with what is measured from the innovation covariances when real observations are compared to the forecast of a

Interactive
comment

model.

p.15, section 4.3.1: as I understand it, an ensemble of forecasts obtained from the UM is used to define the balance operators. To what extent can we expect those to reflect the balance of ABC forecasts?

p.23, eq.(33): given that only a wind potential is used, the multivariate B of eq.(33) needs some explaining.

3. REFERENCES

Bouttier, F., J. Derber and M. Fisher, 1997: The 1997 revision of the Jb term in 3D/4D Var. ECMWF Tech Memorandum No. 238, 54 pages.

Buehner, M., 2005: Ensemble-derived stationary and flow-dependent background-error covariances: evaluation in a quasi-operational setting. Quart. J.R. Meteor. Soc., 131, 1013-1043.

Gauthier, P., M. Buehner and L. Fillion, 1998: Background-error statistics modeling in a 3D variational data assimilation scheme: estimation of the impact on analyses. In Proceedings of ECMWF workshop on diagnosis of data assimilation systems, Reading UK, 2 to 4 November 1998.

---

## Author Comment (AC1) · 9 Jun 2020

**Author response to anonymous referee #1**

Thank you to referee #1 for reading the manuscript and for his/her valuable comments. In the following, the referee's comments are reproduced, and my responses are in blue. Please note that I am instructed by the journal to give responses before preparing a revised manuscript, but I highlight here any changes that I plan to make in the revision.

This paper describes a DA system that has been built for the ABC model of Petrie et al. (2017). The entire system, ABC-DA, is described as flexible, configurable, and efficient enough to be run on a personal computer. One of the stated applications of the ABC-DA is to study convective-scale DA. This system could also clearly be used as a teaching aid for DA students. Section 4 gives a tutorial on variation data assimilaton and a practical "how to" for constructing a 3DVar system. Section 4 gives a tutorial on control variable transforms (CVTs) and is also a practical "how to" for developing CVTs. These two sections are well written tutorials which, by themselves, could serve as a good teaching aids.

I would recommend the publication of this paper after the author addresses a few minor comments.

1. Figure 7 might be more easily interpreted if the correlation rather covariance were plotted. It is not easy evaluate the importance of the cross-covariances (columns 2 & 3).

---

## Author Response (AR1)

Dear Dr Unterstrasser,

Thank you for inviting me to revise my manuscript.

Please find in this PDF my responses to the reviewers (including those to a short comment from a named commentator), and how I have responded. Also in this PDF is the revised manuscript where the changes made are highlighted with blue text.

I hope that you will find the revised manuscript an improvement and will find it suitable for publication in GMD.

Many thanks.

Yours sincerely,

Ross Bannister

**Author response to anonymous referee #1**

Thank you to referee #1 for reading the manuscript and for his/her valuable comments. In the following, the referee's comments are reproduced, my responses are in blue, and my actions to the paper are in red. This reply is identical to the posted reply to the reviewer on the discussions site except that references are made to the changes now made to the manuscript for this revision.

This paper describes a DA system that has been built for the ABC model of Petrie et al. (2017). The entire system, ABC-DA, is described as flexible, configurable, and efficient enough to be run on a personal computer. One of the stated applications of the ABC-DA is to study convective-scale DA. This system could also clearly be used as a teaching aid for DA students. Section 4 gives a tutorial on variation data assimilaton and a practical "how to" for constructing a 3DVar system. Section 4 gives a tutorial on control variable transforms (CVTs) and is also a practical "how to" for developing CVTs. These two sections are well written tutorials which, by themselves, could serve as a good teaching aids.

I would recommend the publication of this paper after the author addresses a few minor comments.

1. Figure 7 might be more easily interpreted if the correlation rather covariance were plotted. It is not easy evaluate the importance of the cross-covariances (columns 2 & 3).

[Figure]

(a) I have computed the correlations (FYI as above), but a certain amount of information that I wish to show is unfortunately lost in such plots, compared to the covariances (i.e. information on the implied variances as well as the correlation patterns), so I would like to keep the covariances in the paper rather than the correlations, if the reviewer agrees.

(b) On the importance of the cross-correlations, I have added (to the discussion around Fig. 7) an interpretation of these in terms of their effect on the analysis increments. Essentially the cross-correlations show

how the assimilation of a $\tilde{\rho}'$ measurement at the yellow cross position in Fig. 7 affect variables like $r'$ and $b'$.

2. A ensemble of forecast perturbations is used for training/developing the CVT. The author describes this raw ensemble in this way: " We regard the raw covariances as a guide to the 'true' covariances that should ideally be modeled by the CVT.". In principal, the ensemble could be made large enough to provide a very accurate covariance, from which the implied covariances (from CVT choices) could be directly evaluated. Is the model state size too large to create a full rank (or nearly full) for this type of comparison?

    (a) Yes, indeed the model state is too large to create a full rank estimate of the B-matrix. I have slightly reworded the sentence cited above to point out the limitations of the raw covariance plot, "We regard the signals contained in the raw covariances as a rough (row-rank) guide to the covariances that should ideally be modelled by the CVT."

3. Following on comment #2. If the training ensemble is of low rank, then it is well known that the covariances must be localized. It is possible that the covariance used for the experiments (i.e. bottom row of Figure 7) should be localized before used for developing the CVT?

    (a) This is true, but the raw covariances still contain a signal that is useful (see my reply above).

    (b) Localisation is not required to develop the CVT though. I have added a new appendix (new appendix A) to show that the information contained in the calibration population is more than enough to determine the CVT. Briefly, there are about $10^5$ pieces of information of the covariance model that need to be determined during the calibration (things like vertical modes, and spectra), but $28 \times 10^6$ pieces of information is provided in the form of 260 super-ensemble members.

4. There is much current research on using ensembles to represent the B matrix in variational DA. Can you comment on any plans to incorporate the ability to directly use an ensemble to perform the background covariance multiply, or possibility a hybrid approach?

    (a) This is something that certainly could be done. I have no plans at present, but I have expanded a bit the path to this at the end of the summary of the paper.
* * *
**Author response to anonymous referee #2**

Thank you to referee #2 for reading the manuscript and for his/her valuable comments. In the following, the referee's comments are reproduced, my responses are in blue, and my actions to the paper are in red. This reply is identical to the posted reply to the reviewer on the discussions site except that references are made to the changes now made to the manuscript for this revision.

Review of the GMD submitted article referenced as: Title: The "ABC-DA system" (v1.4): a variational data assimilation system for convective scale assimilation research with a study of the impact of a balance constraint Author(s): Ross Noel Bannister MS No.: gmd-2019-318 MS Type: Model description paper Iteration: Minor Revision

The paper presents a very comprehensive technical description of a toy data assimilation system based on the previously published ABC-model formulation and codes. Its content is extremely detailed and can easily serve as useful introduction to any scientist, including young scientists and post-graduate students, interested in uploading and using the codes for research or education. The technical content is complemented by a showcase example of the scientific use of the system (the study of the impact of the ABC-associated geostrophic balance constraint in the B-matrix model).

The article is clearly written and the figures are of good quality. The presentation as a whole matches the expected goal which is to provide a scientific introduction to the ABC-DA system. As a reviewer of this article, I do not consider that my role is to evaluate or critizise whether the ABC-model and DA formulation will be a useful scientific tool per se. The present paper will however enable the scientific data assimilation community evaluate that usefulness in practice. I therefore recommend the paper for publication after very minor revisions.

Hereafter follow my minor comments:

1. about §4.1: from what platform can a user upload UM data for initializing the very first steps? Or are such data available with the ABC system packages?
   Ensembles play an important role in many cases of the toy applications. Can you say a few words about how the size of the super-ensemble is set, and what would be a "reasonable" limit of size?

   (a) The size of the sample UM data is unfortunately too large for the GitHub repository. I have though added the sample UM data (and also some pre-prepared ABC ensemble initial conditions) to the web site of the Data Assimilation Research Centre at Reading and provided a link to there from the GitHub repository. Users downloading the ABC-DA system from GitHub now see a URL to the data.

   (b) The second point overlaps with a comment from another reviewer. A new appendix (appendix A) has been added which deals with the amount of information needed to determine aspects of the covariance model (CVT). Essentially there are about $10^5$ pieces of information of the covariance model that need to be determined during the calibration (things like vertical modes, and spectra), but $28 \times 10^6$ pieces of information is provided in the form of 260 super-ensemble members. This is more than adequate.

2. About Fig 6b: this particular plot is actually little discussed in the core text. My question is, noting that the vertical lengthscale increases with increasing vertical mode (i.e. the more nodes on the vertical, the deeper the penetration scale of the mode), is this behaviour due to the fact that the plot holds for the unbalanced part of the scaled density? i.e. one expects the opposite property for the balanced part of scaled density (low-order vertical modes of balanced scaled density would have the largest vertical lengthscales?). Is this correct? I suggest a short comment about Fig 6b could be added in §4.3 or §4.4.

   (a) I actually find that, for the vertical modes, the more nodes there are, the shorter the penetration scale. For info, a low mode of unbalanced scaled density is shown on the left below (short vertical scale) and a high mode is shown on the right (long vertical scale). They appear in this order because the eigensolver evidently outputs modes in ascending eigenvalue order. Such a correspondence between mode and vertical lengthscale is in fact found to be shared with vertical modes of streamfunction (which also share their vertical modes with the balanced component of scaled density). A comment has been added to the penultimate paragraph of Sect. 4.3.5 to state the above correspondence between the number of nodes and the vertical scale, and that the same (not the opposite) property exists for the balanced part of scaled density.

[Figure]

3. about §4.7: it is stated that the ABC-DA system is flexible enough to host a variety of DA methods, like 4D-VAR or Ensemble-Variational formulations. It seems indeed clear from the article that methods based on variational formulations, including iterative steps such as a minimisation and the computation of a gradient, are allowed. However, what about methods like Extended or Ensemble Kalman filters, or versions of Optimal Interpolation, i.e. methods where the Gain Matrix (G) would be somehow explicitly computed, and a direct inversion step involving G would be implied? Similarly, what about methods involving a number of computational steps in observation space as for LETKF (Local Ensemble Transform Kalman filters)? Can the author elaborate in only a few lines on these algorithms, in order to provide an insider view about how easy/how difficult/how different the implementation of such methods in ABC-DA would be?

   (a) As I see it, the extended or ensemble Kalman filters (of whatever flavour), or any of the methods that compute an explicit gain matrix would not follow from such a variational method presented. Some further

comments about possible developments have been added at the end of the summary section concerning the possible extension of the system to ensemble-variational or hybrid systems, including the hybrid gain system of [1].

4. §6, line 776: typo "... that that ..." => "... that the ..."

  (a) This has been corrected.

5. line 774-780: in the discussion of the "control-ability" of the v-component of the wind fields. Is this weak control-ability due to the specific formulation of the ABC toy model? (my guess is "yes"). Can you comment this more in the discussion?

  (a) I think that I was wrong to suggest that $v$ is not controllable in the DA experiments. Instead I need to say that $v$ is updated in the wrong way. I have added more in the summary (and removed the idea that it is due to low controlability). I suggest the the use of observations not at the analysis time (and hence the approximations due to 3DFGAT) might be a reason why the results for $v$ are not good, although this is not proven.

6. 6) §4.2-4.3-4.4 & §5: One general question I have is whether the ABC-DA system can allow the use of a full, total field, B-matrix (that is, one without any balance modeling)? If a total-field B-matrix would be feasible, then the corresponding ABC-DA system could be a valuable reference system for impact studies on specific B-modeling. Can you comment on this?

  (a) I discuss in the new appendix (appendix A) that the number of pieces of information needed to determine a full B-matrix empirically is not really feasible.
* * *
**Author response to short comment from named referee**

Thank you to Pierre Gauthier for reading the manuscript and for his valuable comments. In the following, the referee's comments are reproduced, my responses are in blue, and my actions to the paper are in red. This reply is identical to the posted reply to the reviewer on the discussions site except that references are made to the changes now made to the manuscript for this revision.

Review of paper The ABC-DA system (version 1.4) : a variational data assimilation system for convective scale assimilation research with a study of the impact of a balance constraint By Ross Noël Bannister Submitted to Geoscientific Model Development Date:

**1. MAIN COMMENTS**

The resolution of operational NWP models increases and is reaching the convective scale. This raises new issues associated with the inclusion of ever increasing and complex processes. Moreover, this raises new questions regarding data assimilation methods to be used. Petrie et al. (2017) introduced a simple non-hydrostatic model that could even represent explicitly acoustic waves for data assimilation studies. This paper presents a companion variational data assimilation for this model. The paper is well written and focuses for the most part on the modeling of background error covariance. The emphasis is mostly on 3D-var including the so called FGAT (first-guess at appropriate time) which is also used in many operational systems.

In research, one would like to be able to explore different approaches to assess what could be the best one for the assimilation at the convective scale. I was expecting then that the 4D assimilation would have been more prominent, including 4D-Var in strong and weak constraint for example. In the EnVar, the control variable is the complete model trajectory including temporal correlations for the treatment of background error, which is also used in the weak constraint 4D-Var.

How does this approach compares to using the OOPS/JEDI paradigm which can be used with either a "toy " model or an operational one? I think that it would be important for a "community" model that the advantages of ABC-DA be presented from that perspective. Little is said about observation operators which can easily be the dominant component of a DA system. Assimilating large volumes of Doppler radar data is one example. The point I am making is the paper should make an effort to emphasize aspects of the ABC-DA that could entice researchers to use it.

- As I understand it, the JEDI programme is meant to be a resource to share components of data assimilation systems using an object orientated software approach. The ABC-DA system is not generic, nor object orientated as I understand it, but is suited specifically to the ABC model.

- The possible use of observation operators for Doppler has been added to the abstract and to the introduction.

- The possible development of 4DVar has been mentioned in the summary. For this paper though the focus is on background error covariances (as the reviewer mentions below), for which 3DVar is an adequate tool.

As stated line 143, "much of the design of ABC-DA is concerned with how B is modelled". The emphasis is mostly on multivariate 3D-var including the so called FGAT (first-guess at appropriate time) which is also used in most operational system. Bouttier et al. (1997) introduced balance operators obtained from multilinear regression for a covariance model based on homogenous and isotropic correlations for the analysis variables deemed to have uncorrelated error. This corresponds to some extent to what the paper presents in too much details in my view. I do not see the point of explicitly describing how to use the code: this is important but very likely to change with time. This should be included in a user manual or website for users. On the other hand, it would be more important to explain the scientific justifications. Balance appropriate for the convective scale should be discussed: geostrophic balance does not seem to be the most relevant.

- I have provided what I believe to be enough detail for the paper to be in the spirit of the journal. A separate user guide for the code itself exists via GitHub and is referenced frequently in the paper. I hope the reviewer thinks that this is reasonable.

- Geostrophic balance is still present at the larger scales in the system, which are still analysed here. Geostrophic balance is also used, for instance, in the Met Office's variational DA system for their 1.5km grid size model called UKV, so its relevance is worth studying in this paper. Alternative balance relationships are possible, such as the diagnostic relationship given in Sect. 4.3 of [2], which will be explored in future work.

- These scientific justification points (and details about alternative balances) have been made at the end of Sect. 4.2.1 and in the summary.

The paper ends by presenting the results of experiments to illustrate the impact of the different components representing the balance for this particular model. Is the paper about presenting the ABC-DA emphasizing the advantages of the design to study different aspects of DA that may be important for the assimilation at the convective scale? As it is, most of it is to describe what has been implemented for this particular model of B with some results indicating the impact this may have on the analysis.

I recommend that the author reviews his paper to either present ABC-DA as a polyvalent system for research on DA at the convective scale. Or that it is about a multivariate model for B and its impact on the analysis and forecast. Given the large body of literature on this topic, the latter would be rather thin.

- The ABC model and DA system have been developed over the years specifically to explore B-matrix modelling options. I guess this has had an unconscious influence on the emphasis of this paper. I have though mentioned other possible uses of the system in the introduction (e.g. to explore Doppler observation networks).

The next section presents some specific comments on some, but not all, issues with the paper.

**2. SPECIFIC COMMENTS**

**2.1 Modelling B**

- sections 3.4, 4.2, 4.3, 4.4 (∼14 pages) As stated line 143, "much of the design of ABC-DA is concerned with how B is modelled". Sections The emphasis is mostly on 3D-var including the so called FGAT (first-guess at appropriate time) which is also used in most operational system. Early on, it has been recognized that multivariate covariances should embed dynamical constraints such as an approximate geostrophic balance (e.g., see Daley ,1991). Modeling of a "static" B has been the object of many papers that should be referred to Parrish and Derber (1992) presented the first implementation of 3D-Var and introduced a new approach in which the error was divided into balanced and unbalanced components. They used an ensemble of lagged forecasts at 24 and 48-h to represent averaged background error covariances. Bouttier et al. (1997) introduced balance operators obtained from multilinear regression for a covariance model based on homogenous and isotropic correlations for the analysis variables deemed to have uncorrelated error. This is pretty much what

is presented in section 4.2. I do not see why the author presents this with so much detail given that this is at best, an example of what could be used in the ABC-DA. Buehner (2005) presents a B based on a EOF representation for stationary covariances that can capture some local effects (e.g., presence of orography).

– Parrish and Derber has been referenced towards the end of Sect. 3.4. Derber and Bouttier (1999) has been referenced towards the end of Sect. 4.2.1 (I understand that this is the peer reviewed version of Bouttier et al (1997) that the reviewer cites).

– I have tried to provide as much detail as is necessary to adequately describe the DA system.

**2.2 Other comments**

- p.7: section 3.3. devotes 7 lines to observations. Later, section 5 gives 8 more lines to the observation operators. This is a bit short in my view.

  – Additionally Sect. 4.5 is dedicated to observation operators. I do believe that an adequate discussion of the observation specifications has been made for this particular paper. More technical aspects of the observation network specifications is provided in the technical documentation, which is cited throughout the paper, and available on GitHub. Further mentions of how the system can be extended to include Doppler wind observations have been made (e.g. in the abstract) to help interest readers who may wish to use the ABC system to investigate observation strategies.

- p.6, line 173: The propagator is said to be difficult to derive so it is replaced by the identity. It has been theoretically defined (LeDimet and Talagrand, 1988) and developed for operational model. Even more, the "transpose" of it has also been developed, the "adjoint model". In the context of the incremental form of 4D-Var, some simplifications to the model can be made regarding resolution or the used of a simplified physics. It would be important to know whether it should be possible to expand the ABC-DA to make it possible to do 4D-Var.

  – Reference to LeDimet and Talagrand (1986) has been added to Sect. 3.1 of the paper.

- p.7, line 200: using the analysis variables $\delta(x - x\_b)=$ U with B = UU^T means that we need to get the square root of B but we do not have to invert it. The flip side to this is that if B is singular the increment is built based on the singular vectors of B. For an ensemble like the ETKF, for instance, the increment could only be a linear combination of the members of the ensemble that define B. p.8, lines 207-213: I think this needs to be revised. What is said here only applies to a particular B model with isotropic and homogenous correlations which happens to yield a diagonal matrix when expressed in terms of spectral components (e.g., Fourier, Bessel or spherical harmonics).

  – I have added a short comment here that a singular B-matrix can be modelled using the CVT approach (end of Sect. 3.4). I don't agree with the reviewer that the text applies only to isotropic and homogenous correlations, e.g. if the variances of the spectral components are not just a function of total wavenumber, but of the individual x and y wavenumber components separately, then the covariances can be anisotropic. I have added another short comment (also at the end of Sect. 3.4) that the approach allows more complicated forms of the CVT than $\mathbf{U}_s = \mathbf{\Sigma U}_v \mathbf{U}_h$.

- p.8, line 224: in the incremental form, there is no need to invert U, insofar as the initial point of the minimization is the background state, in which case, initially, $\chi =0$.

  – This is true for the first outer loop where $\mathbf{x}^r = \mathbf{x}^b$, but strictly speaking $\mathbf{U}^{-1}$ is needed for later outer loops when $\mathbf{x}^r \neq \mathbf{x}^b$. This is now mentioned in the text.

- p.8, line 235: the description of the algorithm does not indicate how $\lambda$ is updated.

  – This is given in step 2(f)v.

- p.9, line 245: I agree that the test of transpose is useful to test specific part of the code. But the gradient test (based on a Taylor expansion) should also be mentioned. It is simpler and is routinely used to validate complex operational variational DA systems when al components are active. It should be mentioned.

  – A comment has been added to Sect. 3.6. The gradient test is now coded and included as part of ABC-DA.

- p.10, line 280: I do not think U-1 is needed.

  – It is needed for the calibration. A comment has been added.

- p.10, line 290: the Helmholtz theorem states that there both a potential component and a rotational one (the streamfunction).

  – Equation (13) is for the rotational part and Eq. (14) is for the irrotational part. These equations (and (23) and (24)) are all derived from the Helmholz theorem, $(\delta\mathbf{u}, \delta\mathbf{v}) = \nabla_{\mathrm{h}}\delta\boldsymbol{\chi}_{\mathrm{vp}} + \mathbf{k} \times \nabla_{\mathrm{h}}\delta\boldsymbol{\psi}$ ($\nabla_{\mathrm{h}}$ is the horizontal gradient operator and $\mathbf{k}$ is the vertical unit vector), but simplifications arise due to the lack of latitude dependence in this system. The reader is now reminded of this after Eq. (14).

- p.11, line 296: defining the balance operators is a separate exercise that needs more explanation. Using linear regression has been proposed by Bouttier et al. (1997) and used by others. This requires some insight into the type of balance that could exist. At synoptic scales, geostrophic and Ekman balance have been used to guide the linear regression. What kind of stationary balance can we expect at convective scales. Reference should be made to Parrish and Derber (1992) and Bouttier et al. (1997).

  – Reference to Parrish and Derber has been made at the end of Sect. 3.4 (and Gauthier et al, 1999). The peer-reviewed version of Bouttier et al. (1997) (which is Derber and Bouttier (1999)) has been added at the end of Sect. 4.2.1 which discusses the validity of geostrophic (and hydrostatic) balance. Further possibilities for convective-scales have been discussed in the summary.

- p.12, eq.(23): I think the winds cannot be represented as irrotational and this impacts the very formulation of U. Please revise. This impact the form of B as presented in eq.(22). p.14, line 366: I do not think we can represent B as a finite expansion of its eigenvectors, in general. It is a composition of operators that reflect the general form of the balance operators. In the univariate case for example, with homogeneous and isotropic correlations, this would require a large number of eigenvectors to properly represent it particularly if the characteristic scale is small.

  – The wind does have a rotational part: Eq. (23) is for the irrotational part and Eq. (24) is for the rotational part in this toy model (see also two points above).
  – Only the spatial transforms (vertical and horizontal) are represented as expansions of (approximate) eigenvectors. In the vertical, all 60 eigenvectors are represented (there are 60 levels), and in the horizontal, all 360 wavenumbers are represented (there are 360 longitude points). Hence all scales represented in the model are represented in the DA.

- p.15: section 4.3: one diagnostic used to calibrate the error statistics is to verify if the a priori statistics used in the assimilation are consistent with what is measured from the innovation covariances when real observations are compared to the forecast of a model.

  – A comment that this diagnostic is possible has been added to Sect. 3.6.

- p.15, section 4.3.1: as I understand it, an ensemble of forecasts obtained from the UM is used to define the balance operators. To what extent can we expect those to reflect the balance of ABC forecasts?

  – The UM data are processed to make them consistent with the ABC system, and are then spun-up with an ABC model forecast. I agree that the degree of memory of the UM system has not been tested. A comment has been added to Sect. 4.3.1.

- p.23, eq.(33): given that only a wind potential is used, the multivariate B of eq.(33) needs some explaining.

  – The rotational part of the wind couples to the scaled density field via Eq. (15), which is reflected in (33) (all terms with the $\alpha$ factor).

3. REFERENCES

Bouttier, F., J. Derber and M. Fisher, 1997: The 1997 revision of the Jb term in 3D/4D Var. ECMWF Tech Memorandum No. 238, 54 pages.

[revised manuscript text omitted]

---

## Author Response (AR2)

Thursday 16th July 2020

Dear Dr Unterstrasser,

Many thanks for looking at my revised paper and for requesting minor changes.

Please find the revised version of my paper with the changes made.

As well as adding the section on code availability (including references to the code and data via Zenodo), I have added a section of conflict of interest, plus I have corrected some minor errors to the paper. These concern a missing term in one of the balance equations. The corrections made have no impact on the numerical results as the balance equation in question was not used in the numerical experiments.

The original changes to the manuscript remain in blue text, and the more recent changes appear in red text for your convenience.

Many thanks again for considering my paper.

Yours sincerely,

Ross Bannister